# Primed histone demethylation regulates shoot regenerative competency

Hiroya Ishihara [1,10], Kaoru Sugimoto[1,10], Paul T. Tarr[2], Haruka Temman[1], Satoshi Kadokura[1], Yayoi Inui[1], Takuya Sakamoto[1], Taku Sasaki[3], Mitsuhiro Aida[1], Takamasa Suzuki [4], Soichi Inagaki [5,6,7], Kengo Morohashi[1], Motoaki Seki [3,8], Tetsuji Kakutani[5,7,9], Elliot M. Meyerowitz[2] & Sachihiro Matsunaga [1]

Acquisition of pluripotency by somatic cells is a striking process that enables multicellular organisms to regenerate organs. This process includes silencing of genes to erase original tissue memory and priming of additional cell type specification genes, which are then poised for activation by external signal inputs. Here, through analysis of genome-wide histone modifications and gene expression profiles, we show that a gene priming mechanism involving *LYSINE-SPECIFIC DEMETHYLASE 1-LIKE 3 (LDL3)* specifically eliminates H3K4me2 during formation of the intermediate pluripotent cell mass known as callus derived from *Arabidopsis* root cells. While LDL3-mediated H3K4me2 removal does not immediately affect gene expression, it does facilitate the later activation of genes that act to form shoot progenitors when external cues lead to shoot induction. These results give insights into the role of H3K4 methylation in plants, and into the primed state that provides plant cells with high regenerative competency.

[1] Faculty of Science and Technology, Department of Applied Biological Science, Tokyo University of Science, 2641 Yamazaki, Noda, Chiba 278-8510, Japan. [2] Howard Hughes Medical Institute and Division of Biology and Biological Engineering 156-29, California Institute of Technology, Pasadena, CA 91125, USA. [3] Plant Genomic Network Research Team, RIKEN Center for Sustainable Resource Science, 1-7-22 Suehiro, Tsurumi, Yokohama, Kanagawa 230-0045, Japan. [4] College of Bioscience and Biotechnology, Chubu University, 1200 Matsumoto-cho, Kasugai, Aichi 487-8501, Japan. [5] National Institute of Genetics, 1111 Yata, Mishima, Shizuoka 411-8540, Japan. [6] PREST, Japan Science and Technology Agency, 4-1-8, Honcho, Kawaguchi, Saitama 332-0012, Japan. [7] Department of Genetics, School of Life science, The Graduate University for Advanced Studies (SOKENDAI), Mishima, Shizuoka 411-8540, Japan. [8] Plant Epigenome Regulation Laboratory, RIKEN Cluster for Pioneering Research, 2-1 Hirosawa, Wako, Saitama 351-0198, Japan. [9] Department of Biological Sciences, Graduate School of Science, The University of Tokyo, Hongo, Bunkyo-ku, Tokyo 113-0033, Japan. [10]These authors contributed equally: Hiroya Ishihara, Kaoru Sugimoto. Correspondence and requests for materials should be addressed to K.S. (email: kaoru.sugimoto.lab@gmail.com) or to S.M. (email: sachi@rs.tus.ac.jp)

Acquisition of pluripotency by somatic cells in multicellular organisms is achieved by the removal of epigenetic memories of the original cells and re-establishment of the transcriptional and epigenetic landscape that enables the cells to re-differentiate[1,2]. Coordinated action of both transcriptional and epigenetic factors occur step by step in these processes such that a transcription factor activates/recruits/evicts epigenetic modifiers and epigenetic regulation, in turn, alters chromatin states with high impact on transcriptional output[1–6]. Stem cell studies in animals have described the cellular reprogramming process towards pluripotency as a release of lineage restriction, in which tissue-specific genes become poised for activation in response to external signals[7,8]. The poised states of the genes are maintained and transmitted through different stages of differentiation and facilitate the activation of genes once the appropriate factor profiles are encountered. Gene priming is one of the mechanisms that provides genes with a poised state in advance of actual gene activation.

In mammalian pluripotent cells, such as embryonic stem (ES) cells and induced pluripotent stem (iPS) cells, several types of epigenetic regulation are reported to act in gene priming. They can be broadly divided into two groups. The first includes histone and DNA base modification mechanisms where active or repressive monovalent histone modifications or lack of repressive DNA methylation are stable and reflected in patterns of gene expression at the following stages[7]. Additionally, a bivalent chromatin state with both active and repressive histone modifications is well known as a priming mechanism[7–9]. Tissue-specific genes with these bivalent modifications are expressed only at a low level, or show no detectable expression, in pluripotent cells[8–10]. These genes are poised for immediate activation or repression by loss of the repressive or activating mark, respectively, upon induction of differentiation. The second form of epigenetic regulation includes sequence-specific targeting of gene priming initiated by pluripotency transcription factors[7,11]. In ES cells, pluripotency-related transcription factors are bound to promoters or enhancers of silent tissue-specific genes. As the cells differentiate, the transcription factors are replaced by closely related factors that bind to the same consensus sequences, ultimately contributing to gene activation[12–14]. These different strategies are taken in different contexts of stem cell differentiation and a variety of epigenetic modifications and transcription factors have been found to contribute to gene-priming events in animals[7–9,15].

In plants, however, priming of cell-type-specific genes in pluripotent cells has not been explored in detail, despite the high developmental plasticity of plant tissues. In mammals, somatic cells are reprogrammed into pluripotency through drastic artificial means, such as nuclear transfer into enucleated oocytes or the ectopic expression of pluripotent transcription factors[1]. In contrast, many plants can acquire regenerative competency simply by external signal inputs, such as phytohormone changes and stress applications[16]. These differences suggest that the release of lineage restriction might be easier in plant cells than in animal cells, and that gene-priming systems that facilitate rapid response of cells to external signals might be prevalent in plant regeneration.

One of the key processes for the acquisition of regenerative competency in plant cells is the formation of an intermediate structure of pluripotent cells called callus. In many plant species and tissue types de novo organogenesis is only achieved from pre-induced callus, and not directly from native tissues[17,18]. Regeneration competence is thought to be acquired step-wise with a period of callus growth, which is a prerequisite to organ formation[19]. Therefore, studying callus formation is a useful path toward learning how plant cells acquire and maintain pluripotency. Furthermore, lessons learned from plant callus will provide clues toward enhancement of regenerative ability in recalcitrant species, many of which are important for agricultural science and forestry.

Using a dicot model plant, *Arabidopsis thaliana* (*Arabidopsis*), we previously described the nature of callus induced during in vitro shoot regeneration[20], in which callus is induced from a small piece of plant tissue (explant) on auxin-rich callus-inducing medium (CIM); subsequently, de novo shoots are induced in callus tissues on cytokinin-rich shoot-inducing medium (SIM)[21]. We showed that the callus induced in this regeneration system largely has the identity of root meristem, regardless of its tissue of origin[20,22]; moreover, this root identity is crucial for shoot regenerative competency[23–26]. We also showed that the acquisition of shoot regenerative competency (i.e., the ability to form shoot progenitor cells upon SIM treatment) and the initiation of shoot fate (i.e., the outgrowth of shoots from shoot progenitor cells afterwards on SIM) are two separable processes, which are regulated by different combinations of transcription factors[23]. However, it is not known how or if these processes are regulated at the epigenetic level. Few reports have described the epigenetic basis of the acquisition of shoot regenerative competency, and these studies were limited to specific loci for key shoot and leaf developmental genes and hormone signaling regulators[24–27].

Herein, we reveal a gene-priming mechanism that regulates acquisition of shoot regenerative competency in *Arabidopsis*. We demonstrate that *Arabidopsis LYSINE-SPECIFIC DEMETHYLASE 1-LIKE 3 (LDL3)* specifically demethylates dimethylated lysine 4 of histone H3 (H3K4me2) during callus formation, thereby poising genes for activation in response to subsequent shoot induction. The reduced level of H3K4me2 caused by LDL3, possibly in cooperation with other H3K4me marks, allows activation of genes necessary for acquisition of shoot traits, rather than allowing repression of genes that cancel the original root pattern of gene expression in callus tissue. In addition, we identify LDL3 target genes and demonstrate that they are involved in the initial process of shoot induction. Thus, the primed histone demethylation regulated by *LDL3* is an epigenetic mechanism that provides plant cells with shoot regenerative competency during callus formation.

## Results

**LDL3 plays a predominant role in de novo shoot regeneration.** To gain insight into the epigenetic modifications in shoot regeneration, we focused on epigenetic regulators whose expression is up-regulated during callus formation[20] and carried out screening of mutant plants deficient in those genes for their regenerative capability. We found that T-DNA insertion alleles for *LDL3*, one of the *Arabidopsis* orthologs of human *LYSINE-SPECIFIC DEMETHYLASE 1 (LSD1)* (also known as *KDM1*), are not competent to regenerate shoots (Fig. 1a, b; *ldl3-1* and *ldl3-2*). LSD1 is evolutionarily conserved among eukaryotes and removes mono- and dimethyl groups from H3K4[28–30]. In mammals, LSD1 is involved in a broad spectrum of developmental processes and disease, such as embryonic pluripotency, cellular differentiation, and cancer initiation and growth[31]. In *Arabidopsis*, four LSD1 paralogs are found: flowering locus D (FLD), LDL1, LDL2, and LDL3[32–35] (Fig. 1c). The first three show extensive similarity with each other, and promote flowering by suppressing key floral repressors via the reduction of H3K4me[32–34]. LDL3 shows less similarity to the other paralogs[33], and no studies of its role in plant development have been reported.

While shoot regeneration was dramatically reduced in *ldl3-1* and *ldl3-2* root explants compared with wild-type explants, callus formation of those mutants was not affected (Fig. 1a, b). The shoot regenerative ability of *ldl3-1* (hereafter referred to as *ldl3*)

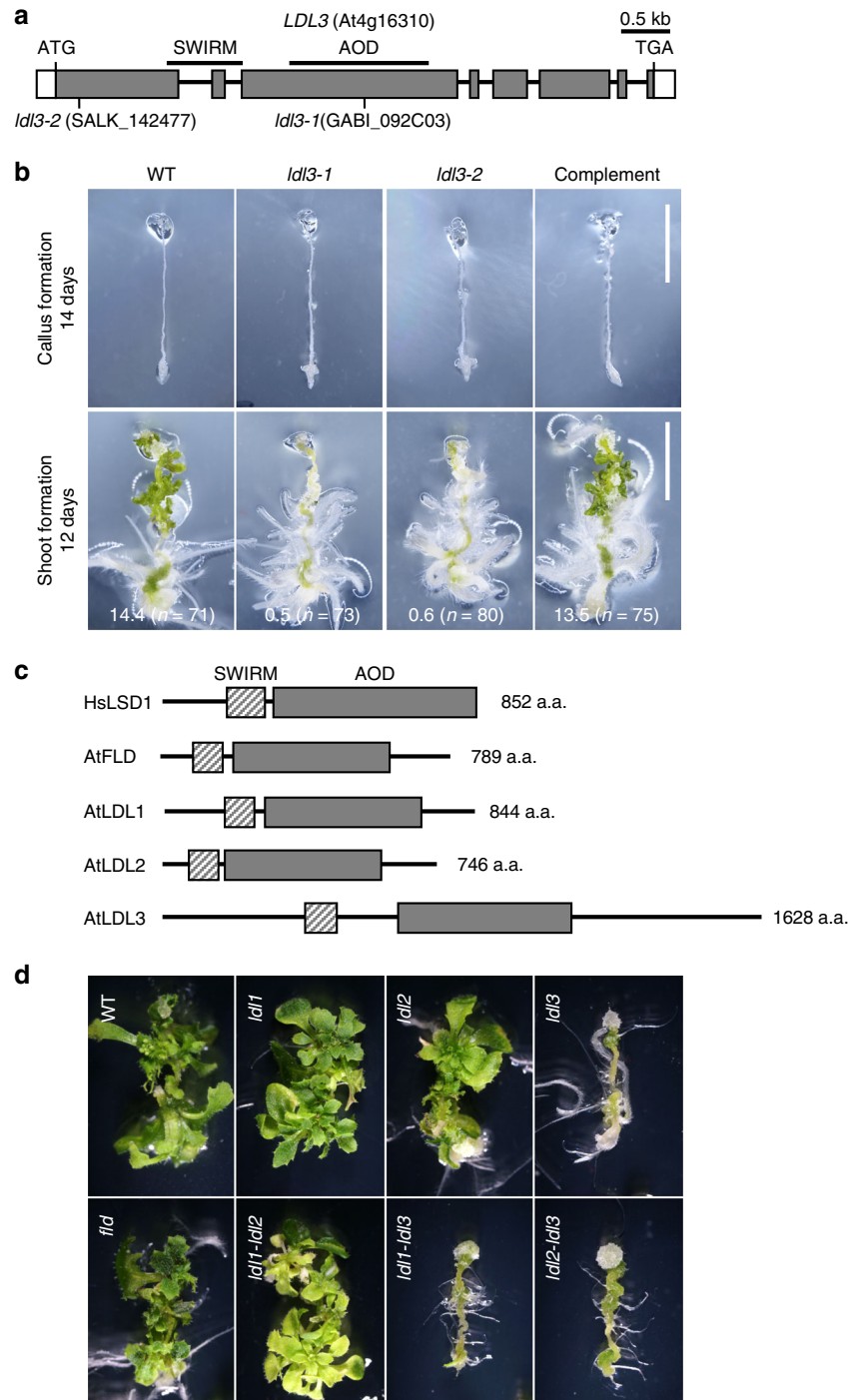

**Fig. 1** LDL3 is required for de novo shoot regeneration. **a** Structure and sites of T-DNA insertion in the *LDL3* genes. Boxes, exons; bars, introns; SWIRM, predicted chromatin binding domain; AOD, amine oxidase domain including demethylase catalytic center[28]. **b** Phenotype in shoot regeneration. Root tip explants were excised from seedlings of wild type, *ldl3* mutants, and a complementation line (*pLDL3::LDL3-GFP* in an *ldl3-1* homozygous background) at 6 days after germination, and incubated on CIM for 14 days and on SIM for 12 days. A visible apical meristem surrounded by 2–3 leaves with trichomes was counted as one shoot in each explant (values are mean ± s.d. WT, 14.4 ± 4.0 s.d., n = 71; *ldl3-1*, 0.5 ± 0.9 s.d., n = 73; *ldl3-2*, 0.6 ± 1.4 s.d., n = 80; Complement: 13.5 ± 4.3 s.d., n = 75). **c** Structure of human LSD1 (HsLSD1) and its *Arabidopsis* paralogs (AtFLD and AtLDL1–3). LDL3 shows less similarity to the other proteins. **d** Shoot regeneration phenotype in single or double *LDL* gene mutants. The shoot regeneration rate was reduced only in mutants for *LDL3*. Scale bar: 5 mm. See also Supplementary Fig. 1

was recovered by the expression of LDL3-GFP under its own promoter (*pLDL3::LDL3-GFP*) (Fig. 1b). We also examined single and double mutants for *LDL* genes, and found that shoot regeneration was only suppressed in mutants deficient in *LDL3*

(Fig. 1d), indicating that among the paralogs, *LDL3* is a major contributor to de novo shoot regeneration. The *ldl3* mutant did not display a strong phenotype in plant development. There was no significant difference between wild type and *ldl3* mutants in

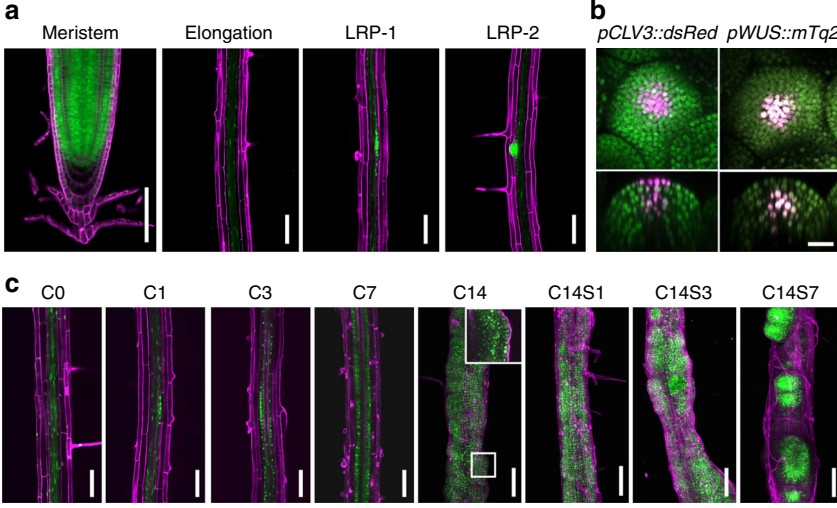

**Fig. 2** *LDL3* is expressed in the plant meristems and callus tissue. The expression pattern of the LDL3 translational reporter (green) in untreated roots. Meristem, a root apical meristem; Elongation, an elongation region; LRP-1, an early lateral root primordium; LRP-2, a late lateral primordium. (**a**), in SAM with expression of the SAM markers *pCLV3::dsRed* and *pWUS::mTq2* (magenta) (**b**), and in root explants undergoing regeneration processes (**c**). All panels in (**a**), lower panels in (**b**), and the panels of C0–7 and the inset of C14 in (**c**) are single optical sections; all other panels are projections from confocal Z-stacks. The panels of C14—C14S7 in (**c**) are combined images of two sequential projections. Cellular outlines were visualized with PI staining (magenta) (**a** and **c**). Scale bars: 100 μm (**a** and **c**) and 20 μm (**b**). See also Supplementary Fig. 2

either the size of the shoot apical meristem (SAM), or root apical meristem (RAM), or in root length (Supplementary Fig. 1).

**LDL3 is expressed in plant meristems and callus tissue**. We next examined the spatio-temporal expression pattern of *LDL3* in plants and explants using the transcriptional and translational reporters *pLDL3::GUS* and *pLDL3::LDL3-GFP* (Fig. 2, Supplementary Fig. 2). *pLDL3::GUS* signal was detected in vascular bundles and meristems such as RAM, SAM, and lateral root primordia (LRP) (Supplementary Fig. 2a). The reporter was highly expressed throughout callus tissue. LDL3 protein localization was visualized by *pLDL3::LDL3-GFP* at the cellular level (Fig. 2). In the root, the reporter was weakly expressed in vascular tissues and strongly expressed in the LRP and RAM tissues, except the endodermis and quiescent center (QC) regions (Fig. 2a). In the shoot, it was expressed throughout the entire SAM region, overlapping with the expression domains of SAM-expressed genes *WUSCHEL* (*WUS*) and *CLAVATA3* (*CLV3*)[24,36–38] (Fig. 2b). During callus formation, it was highly expressed in the growing regions but excluded from the sub-epidermal layer of callus tissue, which was previously reported to express QC markers[23] (Fig. 2c; CIM 14 days (C14)). Upon shoot induction, the signal gradually localized to shoot progenitor cells that are starting to form a meristem (Fig. 2c; SIM 3 days and 7 days (C14S3 and C14S7)). Consistent with the expression pattern of the reporter, *LDL3* transcription was up-regulated during callus formation and at the initial stage of shoot induction (Supplementary Fig. 2b).

**LDL3 allows gene activation for acquisition of shoot trait**. Because the callus induced in this system is similar to root meristem, shoot regeneration from callus tissue is thought to involve the trans-differentiation from root-like to shoot patterns of gene expression[20,22,39]. To compare transcriptome changes during this transition in wild type and *ldl3*, we performed RNA sequencing (RNA-seq) in root explants of both lines at CIM 14 days and SIM 1 and 7 days (C14, C14S1, C14S7).

By comparing the transcriptome of wild-type explants before and after shoot induction (C14 vs C14S1 or C14 vs C14S7), we first identified the genes that were up- or downregulated (FC > 1.25 or < 0.8, $p < 0.01$) in response to shoot induction at early or late stages in wild type (named UGs_S1, DGs_S1, UGs_S7, and DGs_S7, respectively), and examined the behavior of these genes in *ldl3*. We found that the up-regulated genes (UGs_S1 and UGs_S7) were less upregulated in *ldl3* than in wild type, at both the early and late stages of shoot induction (Fig. 3a, Supplementary Fig. 3a, b, $p < 0.01$). In contrast, the downregulated genes (DGs_S1 and DGs_S7) were eventually down-regulated in *ldl3*, as well as in wild type (Supplementary Fig. 3c, d, C14S7, $p > 0.05$), although the early down-regulated genes (DGs_S1) were less suppressed in *ldl3* compared with wild type in the initial stage of shoot induction (Supplementary Fig. 3c, C14S1, and Fig. 3c). This suggested that the *ldl3* explant has defects in gene activation in response to shoot inductive conditions. Given that neither tissue morphology nor reporter expression patterns in the wild-type explants are apparently changed during the first day of shoot induction (C14S1) (Supplementary Fig. 4), the failure of gene activation in the *ldl3* mutant might not be merely caused by the absence of tissues that would express these genes, but might also arise from the lack of competence of the mutant to activate the genes that lead to shoot progenitor cell formation in response to shoot induction. In contrast, suppression of genes in *ldl3* occurs normally, albeit slowly at first. We thus speculated that *ldl3* fails to acquire shoot traits but successfully cancels the root gene expression patterns of callus upon shoot induction.

To test this hypothesis, we examined genes expressed in the SAM and RAM. Sixteen out of twenty four SAM genes tested were highly upregulated at C14S7 once shoot tissues were initiated in wild-type explants (FC > 1.25), while only 11 of them were moderately upregulated in *ldl3* (Fig. 3b), which reflects the regeneration phenotype. We also observed the spatial pattern of SAM reporters *pWUS::mTq2* and *pCLV3::dsRed-N7*[40]. In wild type, both reporters were strongly expressed in shoot progenitor cells from C14S5 and were localized in the SAMs formed in the callus at C14S7; in *ldl3*, their expression was rarely detected at these stages (Fig. 3e, f), indicating as does its visible phenotype that *ldl3* is incompetent to regenerate shoot progenitor cells de novo. Almost all RAM genes tested were highly expressed in the callus at C14 and were eventually downregulated after shoot

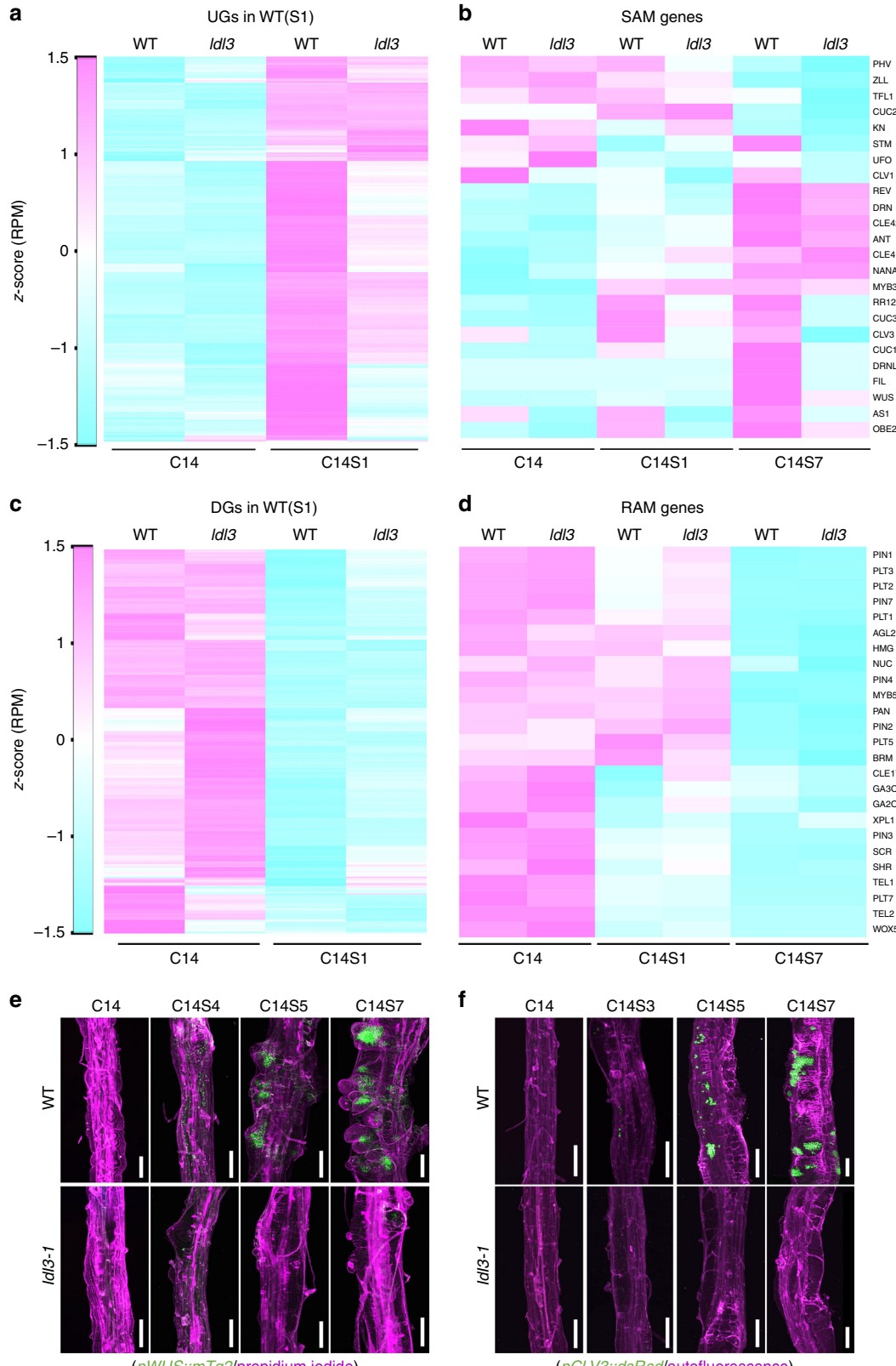

**Fig. 3** LDL3 activates genes for the acquisition of shoot traits upon shoot induction. Comparing transcriptome profiles of wild type (WT) before and after shoot induction (C14 vs C14S1 or C14 vs C14S7) identified genes that are up- or down-regulated (FC > 1.25 or < 0.8, $p < 0.01$) in response to shoot induction at early or late stages in WT (UGs_S1, DGs_S1, UGs_S7, and DGs_S7). **a–d** Expression changes of UGs_S1 (**a**), key developmental genes for SAM (**b**), DGs_S1 (**c**), and key developmental genes for RAM (**d**) upon shoot induction in WT and *ldl3*. Heat maps show relative expression levels for the genes by *z*-scores of read counts per million mapped reads (RPM). **e**, **f** Expression patterns of SAM reporters (green), *pWUS::mTq2* (**e**), and *pCLV3::dsRedN7* (**f**), in calli of WT and *ldl3* before and after shoot induction. All panels are combined images of two sequential projections. Cellular outlines (magenta) were visualized with PI staining (**e**) or autofluorescence (**f**). Scale bars: 100 μm. See also Supplementary Fig. 3, Supplementary Data 3 and 4

induction at C14S7 in *ldl3* as well as in wild type, although their initial down-regulation in *ldl3* was at a lower level than wild type at C14S1 (Fig. 3d). This indicated that *ldl3* successfully acquires and cancels root meristem traits during callus formation and subsequent shoot induction processes.

**LDL3 preferentially removes H3K4me2 during callus formation**. To elucidate the effect of LDL3 on H3K4 methylation in calli, we performed genome-wide chromatin immunoprecipitation followed by sequencing (ChIP-seq)[41] for H3K4me1, H3K4me2, and H3K4me3 in wild type and *ldl3* explants. We observed increased H3K4me2 and H3K4me3 levels in wild-type explants during callus formation (Supplementary Fig. 5a). In *ldl3* explants, H3K4me2 increased even more than in wild type at all stages of the regeneration process (C0, C14, and C14S1), whereas the other modifications were unchanged (Fig. 4a), indicating that LDL3 specifically regulates H3K4me2 during callus formation. We confirmed that many of the genes with hyper H3K4me2 in *ldl3* (4214 out of 6539 genes) overlapped with LDL3-bound genes (6243 genes), which were identified by comparing anti-GFP ChIP peak profiles of calli (C14) derived from *pLDL3::LDL3-GFP* in *ldl3* and *p35S::GFP* in wild type (Col-0) ($q < 0.1$) (Fig. 4b). The positional profiles of H3K4me2 (in wild type and *ldl3*) and LDL3-GFP (in wild type) on the genic region (gene body plus 2 kb sequence up- and downstream) of 6243 LDL3-bound genes showed that LDL3-GFP was predominantly bound from the center towards the 3′ region of the gene body, where H3K4me2 increased in *ldl3* compared with wild type (Fig. 4c). We also observed the reduction of H3K4me2 in this region of LDL3-bound genes in wild-type explants during callus formation, which was not observed in LDL3-unbound genes (Supplementary Fig. 5b). These data demonstrated that LDL3 binds to the gene body and correlates there with removal of H3K4me2, although the total amount of H3K4me2 is increased during callus formation.

To further confirm the demethylase activity of LDL3 in vivo, we next over-expressed *p35S::LDL3-GFP* or *p35S::mLDL3-GFP*, which contains a point mutation (lysine 949 to alanine) in the amino oxidase domain, in tobacco (*Nicotiana benthamiana*) leaf and examined H3K4 methylation by immunofluorescence microscopy[42,43]. The comparison of nuclei with and without GFP signals showed that LDL3-GFP, but not mLDL3-GFP, substantially reduced H3K4me2 (Fig. 4d, e, $p = 2.59E\text{-}39$), moderately reduced H3K4me3 (Supplementary Fig. 6, $p = 8.35E\text{-}06$), and had no significant effect on H3K4me1 (Supplementary Fig. 6, $p > 0.05$). Therefore, LDL3 could demethylate H3K4me2 and H3K4me3 in plant cells. Taken together with the ChIP-seq data, we concluded that LDL3 preferentially removed H3K4me2 in *Arabidopsis* callus formation.

**LDL3 regulates regenerative competency via H3K4me2 removal**. To investigate when LDL3 removes H3K4me2 during shoot regeneration processes, we next compared temporal changes of histone modification profiles of wild type and *ldl3* explants. The increased levels of H3K4me2 during callus formation (C14–C0) in *ldl3* were even higher than in wild type (Fig. 5a), while the changes in H3K4me2 upon shoot induction (C14S1–C14) were subtle in both wild type and *ldl3* (Fig. 5b; black dots). This suggested that LDL3 removes H3K4me2 during callus formation, and the status of H3K4me2 is apparently maintained beyond shoot induction without additional removal. This result was unexpected given the drastic phenotype of *ldl3* in shoot regeneration but not in callus formation, with gene activation only after shoot induction severely perturbed. Focusing on early up- or downregulated genes upon shoot induction (UGs_S1

or DGs_S1), we found no major changes in their H3K4me2 status upon shoot induction in either wild type or *ldl3* (Fig. 5b; red and blue dots).

We then investigated how H3K4me2 and other modifications influence gene expression levels in the explants. In animals and plants, H3K4me3 is known to associate with active gene expression[44–46], while the exact roles of H3K4me1/2 in plants are still unclear[45], although they have often been linked with transcriptional activation in animals[44,47]. Methylation levels of H3K4me2 in wild-type calli (C14) did not correlate with gene expression levels in callus tissues (Supplementary Fig. 7). Among the modifications tested here, only H3K4me3 levels exhibited a significant positive correlation with gene expression levels. We next investigated the relationship between hyper H3K4me2 caused by *ldl3* during callus formation (C14), and the altered gene expression of mutant explants before (C14) and after (C14S1) shoot induction. The changes in H3K4me2 levels in *ldl3* calli (*ldl3*/WT at C14) did not correlate with gene expression changes at the same stage (*ldl3*/WT at C14) but instead with changes during shoot induction (*ldl3*/WT at C14S1), indicating that H3K4me2 levels in calli (C14) primed the expression states found after induction (C14S1) (Fig. 5c–e; groups ii, iii). To further confirm that this effect was caused by H3K4me2 mediated by LDL3, but not H3K4me1 or H3K4me3, we examined the relationship between the altered H3K4 methylations caused by *ldl3* in calli and the altered gene expression of mutant explants before and after shoot induction, focusing on LDL3-bound genes (Supplementary Fig. 8). The H3K4me1 and me3 levels of LDL3-bound genes rarely differed between wild type and *ldl3* calli, only H3K4me2 of these genes was increased in *ldl3* (Supplementary Fig. 8a). Moreover, hyper H3K4me2, but neither hyper-H3K4me1 nor -H3K4me3, at LDL3-bound genes in *ldl3* calli (C14) showed a negative correlation with gene expression changes upon shoot induction (C14S1) but not with those prior to shoot induction (C14) (Supplementary Fig. 8b). Therefore, the results are consistent with LDL3 regulating later activation of its associated genes upon shoot induction (C14S1), but not during callus formation (C14), via H3K4me2 removal.

Because H3K4me2 is widely linked to gene activation in animals, we also examined the possibility that genes with hyper H3K4me2 in *ldl3* contain repressive histone modification as well. As H3K27me3 is often found together with H3K4me2 on bivalent chromatin in the context of stem cell differentiation and gene imprinting in animals[45,48,49], we investigated the state of H3K27me3 during the regeneration process. The numbers of hyper H3K27me3 genes in *ldl3* (*ldl3*/WT) before and after shoot induction (C14 and C14S1) were extremely small compared to that of hyper H3K4me2 genes in *ldl3* calli (*ldl3*/WT), and the genes in these sets hardly overlap (Supplementary Fig. 9a). Moreover, the levels of hyper H3K27me3 on the hyper H3K4me2 genes did not correlate with the degree of gene expression changes (*ldl3*/WT) upon shoot induction (C14S1) (Supplementary Fig. 9b). This suggests the repressive role of hyper H3K4me2 in *ldl3* callus upon shoot induction, without repressive H3K27me3.

These data demonstrated that LDL3 activity is associated with erasure of H3K4me2 during callus formation, which could make the affected genes competent to be activated in response to shoot induction; alternatively, LDL3 could also affect the probability of later shoot induction by a different process, with later shoot induction correlated with H3K4me2 erasure.

**LDL3-mediated H3K4me2 works collaboratively with H3K4me1/3**. As a previous report on *Arabidopsis* seedlings suggested that gene expression levels are associated with different

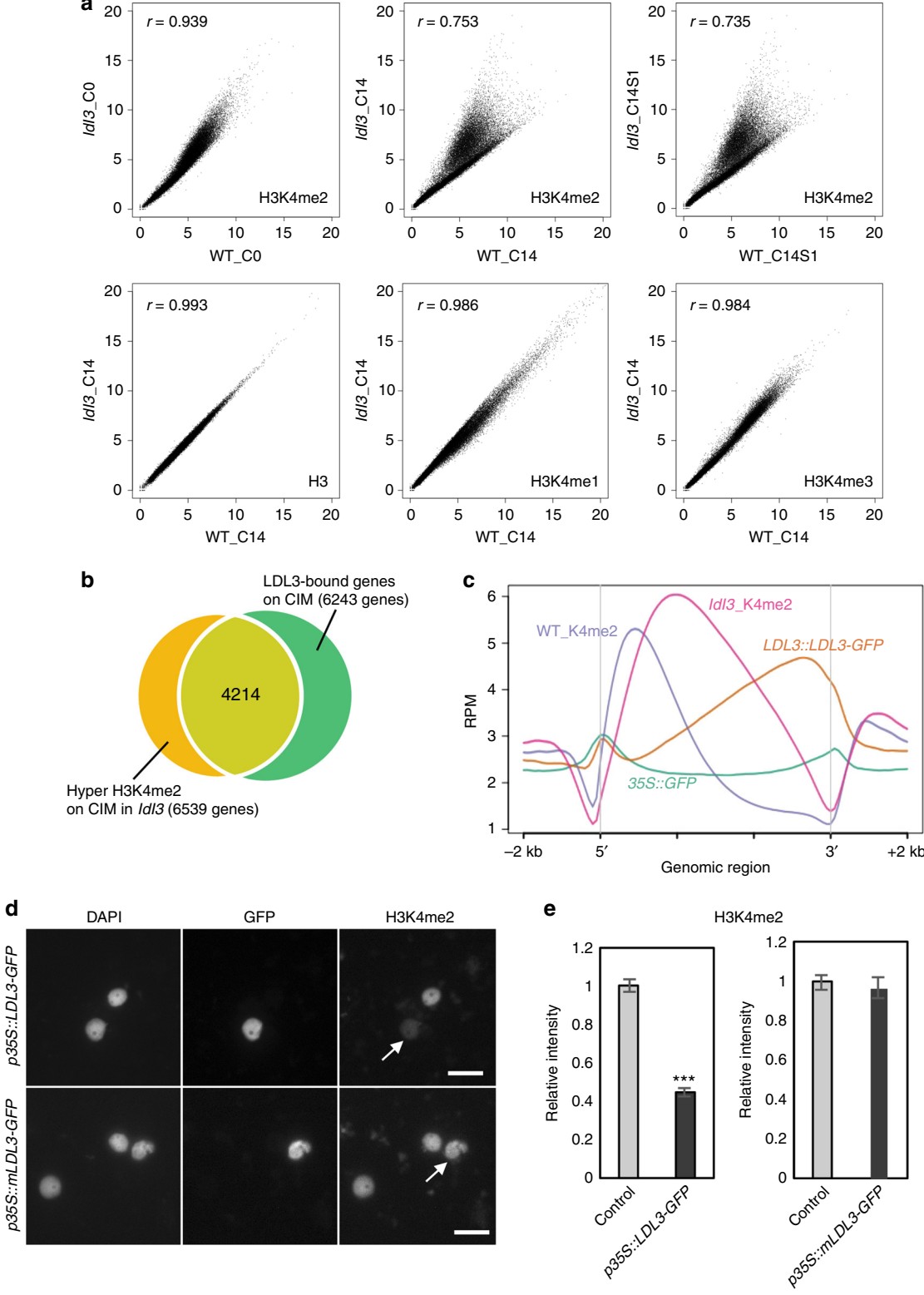

**Fig. 4** LDL3 preferentially regulates the removal of H3K4me2 in CIM. **a** H3 and H3K4 methylation levels in *ldl3* compared with WT at each stage. Each dot represents the square root of RPM. *r*: Pearson correlation coefficient. **b** Venn diagram of the genes with hyper H3K4me2 in *ldl3* (read count_*ldl3*/WT > 1.5, *p* < 0.01) and LDL3-bound genes (identified by comparison of ChIP profiles for GFP in *pLDL3::LDL3-GFP/ldl3* and *p35S::GFP/*WT (*q* < 0.1)) (C14). **c** Positional profiles of H3K4me2 and LDL3 on 6243 LDL3-bound genes. **d** Nuclei transfected with *p35S::LDL3-GFP* or a mutated construct *p35S::mLDL3-GFP* were mixed with control nuclei without transfection. All nuclei were visualized by DAPI staining, and GFP and H3K4me2 were visualized by immunostaining. Arrows indicate nuclei transfected with *p35S::LDL3-GFP* or *p35S::mLDL3-GFP*. Scale bars: 20 μm. **e** Quantification of the immunostaining signals. The transfected nuclei with GFP signal versus non-transfected nuclei without GFP signal (control) were observed. A maximum of five control nuclei per one transfected nucleus were randomly picked up from the same field and assessed for signal intensity. Values are mean ± s.d. ***\*\*\****p* < 0.001 (Student's *t*-test). 132 < *n* < 242 (Control), *n* = 59 (*p35S::LDL3-GFP*), *n* = 34 (*p35S::mLDL3-GFP*). Source data are provided as a Source Data file

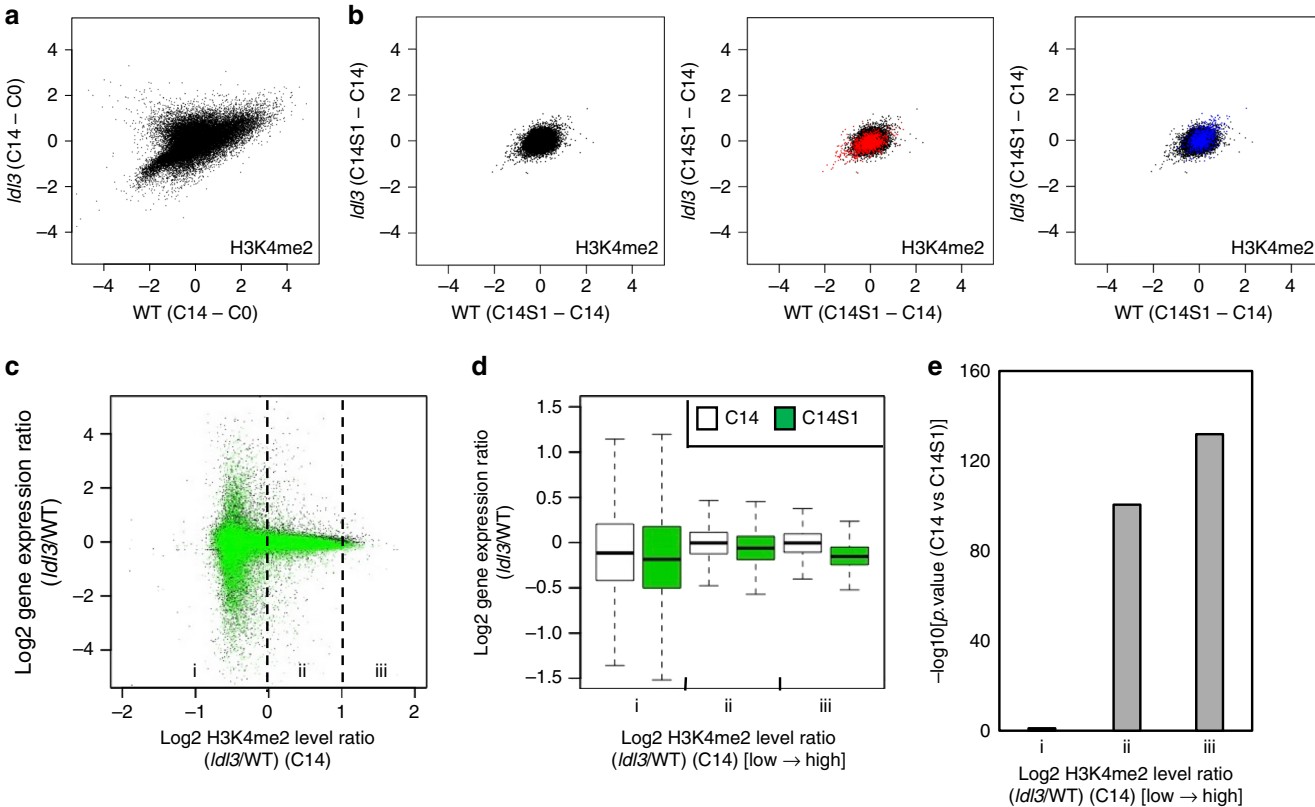

**Fig. 5** LDL3-mediated H3K4me2 in CIM regulates gene expression on SIM. **a** H3K4me2 changes during callus formation (C14–C0) in *ldl3* compared with WT. **b** H3K4me2 changes upon shoot induction (C14S1–C14) in *ldl3* compared with WT. Each dot represents the square root of RPM. Black dots, all genes; red dots, upregulated genes in C14S1 in WT (UGs_S1); blue dots, downregulated genes in C14S1 in WT (DGs_S1). **c** Altered gene expression in *ldl3* [Log2 (read counts per kilobase million mapped reads: RPKM_*ldl3*/WT)] before (C14, black dots) and after (C14S1, green dots) shoot induction with reference to hyper H3K4me2 caused by *ldl3* [Log2 (RPKM_*ldl3*/WT)] during callus formation (C14). **d** On the basis of H3K4me2 changes between WT and *ldl3* [Log2 (RPKM_*ldl3*/WT)] at C14 in (**c**), genes were divided into three groups (*i* < 0, 0 < *ii* < 1, *iii* > 1) (*x*-axis). Gene expression fold-changes between WT and *ldl3* [Log2 (RPKM_*ldl3*/WT)] at C14 and C14S1 were calculated and compared across groups. **e** −log10 [p value (C14 vs C14S1, Wilcoxon signed rank test)] for each group in (**d**). Source data are provided as a Source Data file

assortments of mono (me1$^+$), di (me2$^+$), and tri (me3$^+$)-methylation of H3K4[45], we also assessed expression levels of the genes with different combinations of H3K4me marks in wild-type callus (C14). As previously shown[45], high and low expression of genes is likely to associate with the presence and absence of H3K4me3, respectively, suggestive of the role of H3K4me3 in active transcription (Supplementary Fig. 10), which is consistent with the results described above (Supplementary Fig. 7). Then we investigated whether LDL3-mediated H3K4me2 reduction in calli (C14) regulates gene expression upon shoot induction (C14S1) in association with the states of other H3K4me marks, or independently from them. The Venn diagram of me1$^+$genes, me3$^+$genes, and LDL3-bound genes showed that LDL3-bound genes were most frequently found in the me1$^+$me3$^+$ group (Fig. 6a), suggesting that LDL3 might be recruited mainly to me1$^+$me3$^+$ genes in the callus. This is also consistent with the result that H3K4me1 and H3K4me3 in LDL3-bound genes are increased during callus formation (Supplementary Fig. 5b). We next examined the effects of hyper H3K4me2 on altered gene expression in *ldl3* mutants upon shoot induction in each group of LDL3-bound genes. We found that only the me1$^+$me3$^+$ group of LDL3-bound genes showed a strong negative correlation between the hyper H3K4me2 levels in calli (C14) and the altered gene expression (*ldl3*/WT) upon shoot induction (C14S1), but not prior to shoot induction (C14) (Fig. 6b, c). In fact, 14 out of 16 putative LDL3 target genes selected in the following section

belong to the me1$^+$me3$^+$ group. Thus, the effect of reduced H3K4me2 on gene expression might be associated with the status of other H3K4me marks.

Taken together, the evidence suggests that LDL3 may be preferentially recruited to genes with H3K4me1 and H3K4me3 marks, and LDL3-mediated H3K4me2 removal allows activation of genes upon shoot induction, possibly in cooperation with H3K4me1 and H3K4me3 marks. As for the target selection, however, we cannot exclude other scenarios where the genes are marked at earlier stages of CIM incubation, as we obtained the data from C14 explants. It is also possible that LDL3 might use other marks to find target genes, such as other states of H3K4me or other histone/DNA modifications, and that the m1$^+$m3$^+$ states of H3K4 in the target genes might be a result of the feedback effects of LDL3 activity, which involves other H3K4 methyltransferases or demethylases.

**LDL3 target genes regulate shoot regeneration**. To select putative LDL3 downstream target genes involved in shoot regeneration, we compared the lists of genes from our data. Because LDL3 regulates fate conversion upon shoot induction prior to the formation of shoot progenitor cells, we especially focused on the genes that are upregulated in response to shoot induction at the earliest stage (S1). During the initial day after shoot induction, many SAM genes are not up-regulated and root genes are also not yet completely inhibited (Fig. 3b, d)[50]. Therefore, a distinct shoot or root

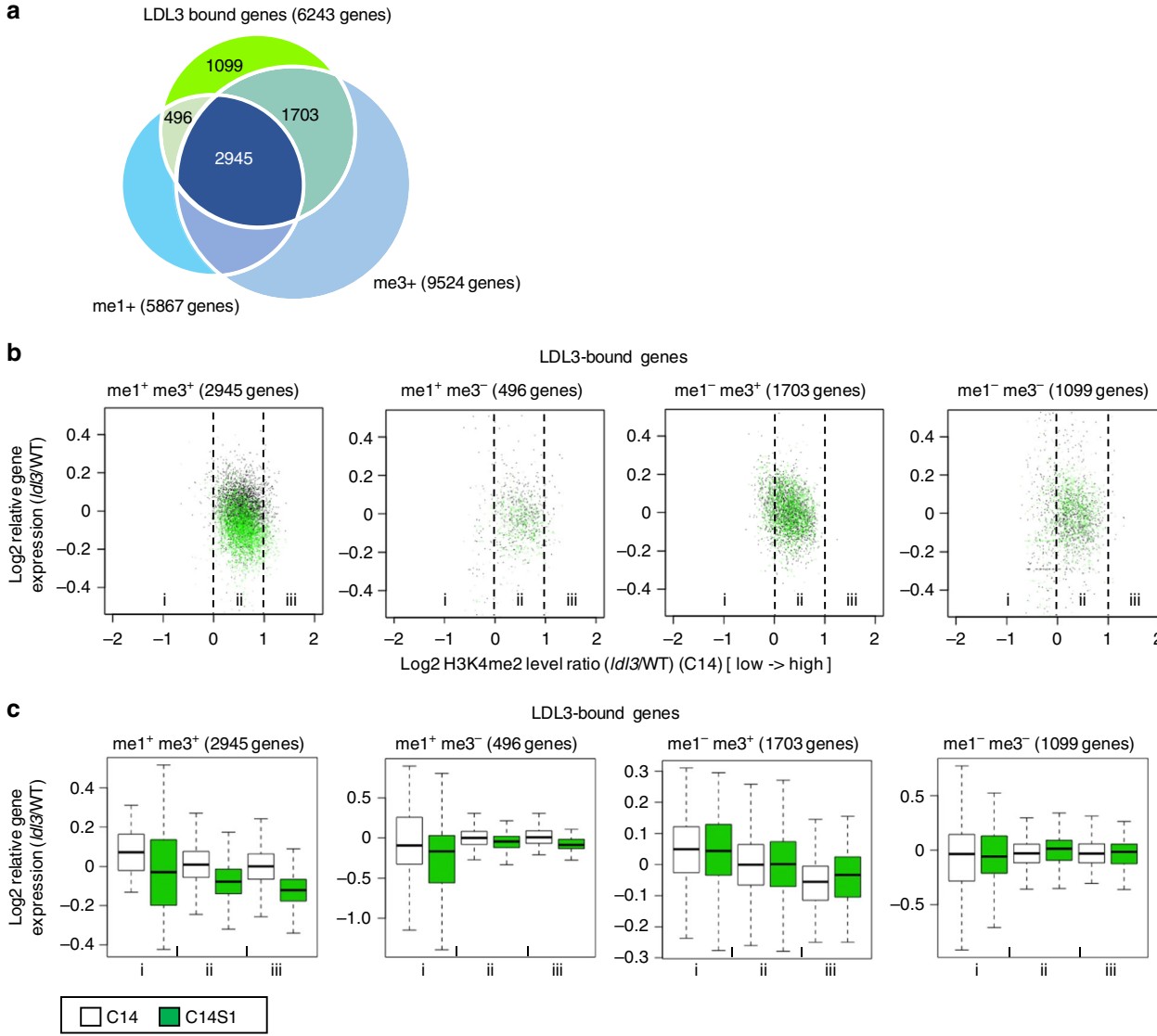

**Fig. 6** LDL3-mediated H3K4me2, in cooperation with other H3K4me marks, regulates gene expression. **a** Venn diagram of LDL3-bound genes with or without either H3K4me1 or H3K4me3 in wild-type callus (C14). LDL3-bound genes were classified into four groups according to the presence or absence of high level of H3K4me1 or H3K4me3 (> 20 RPKM). 2945 genes (47.2%) of LDL3-bound genes were found on me1+me3+genes. **b** Altered gene expression in *ldl3* before (C14, black dots) and after (C14S1, green dots) shoot induction with reference to hyper H3K4me2 caused by *ldl3* in LDL3-bound genes included in four gene sets, assorted by H3K4me1 and H3K4me3 states. **c** Based on the changes of H3K4me2 between WT and *ldl3* [Log2 (RPKM_*ldl3*/WT)] at C14 in (**b**), each set of LDL3-bound genes was divided into three groups (*i* < 0, 0 < *ii* < 1, *iii* > 1). Gene expression fold-changes between WT and *ldl3* [Log2 (RPKM_*ldl3*/WT)] at C14 and C14S1 were calculated and compared across groups. Source data are provided as a Source Data file

identity is not clearly readable at this time point. The molecular events that lead towards shoot fate specification in explants during this narrow window of time are unclear. The intersection of the Venn diagram shown in Fig. 7a revealed 16 genes that are bound and H3K4me2-demethylated by LDL3 during callus formation, and are initially upregulated upon shoot induction in wild type, but not in *ldl3* (Fig. 7a, b). We assessed the regenerative phenotype of mutants for these 16 genes, and found that mutants in *CBL-INTERACTING PROTEIN KINASE 23* (*CIPK23*), *NADH-DEPENDENT GLUTAMATE SYNTHASE 1* (*GLT1*), and *UBIQUITIN-PROTEIN LIGASE 4* (*UPL4*) showed reduction in shoot regeneration, though their phenotypes were moderate compared with *ldl3-1* and *ldl3-2* mutants (Fig. 7c, d). Histone H3K4 methylation and LDL3 binding patterns at these three genes showed that LDL3 binds to the center-to-3′ region of the gene bodies, where H3K4me2 accumulates in *ldl3* compared with wild type (Fig. 7e).

We also identified three genes that are bound and H3K4me2-demethylated by LDL3 among the shoot meristem genes listed in Fig. 3b and Supplementary Fig. 11. Among them, *ARABIDOPSIS RESPONSE REGULATOR 12* (*ARR12*) was downregulated in *ldl3* compared with wild type, which became significant at S7 (S7, $p = 0.00371$, FC = 0.718; S1, $p = 0.02446$, FC = 0.823). Double and triple mutants of *ARR* genes containing the *arr12* mutation previously displayed suppressed shoot regeneration phenotypes[35], so the phenotype of *ldl3* might be attributable to the repression both of early-response genes and *ARR12*.

## Discussion
In *Arabidopsis* as well as in many other plants, many tissues are not ready directly to regenerate shoots de novo but require the formation of callus as an intermediate step. While callus tissue is similar to the root meristem in its transcriptional profile and

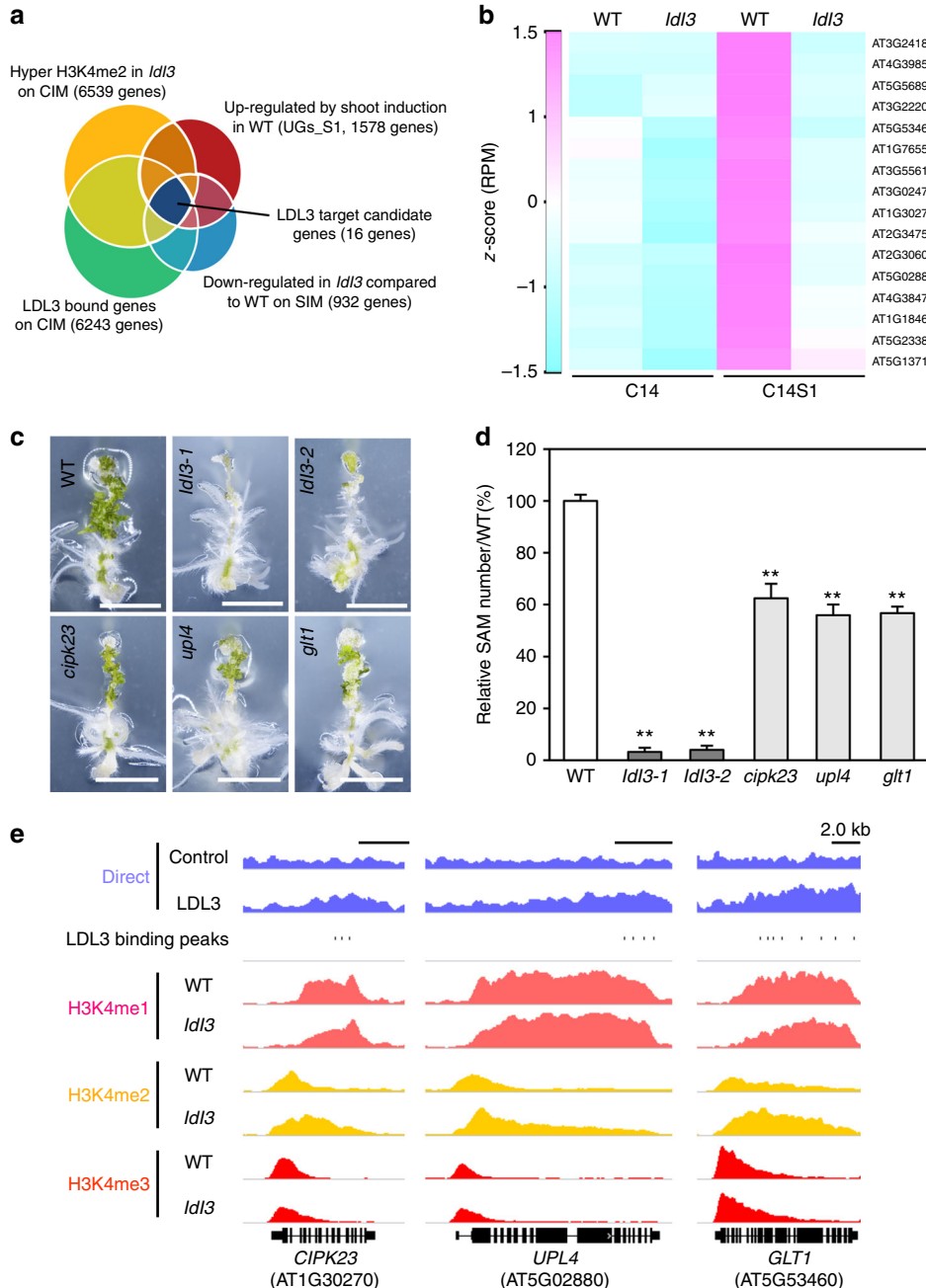

**Fig. 7** Identification of LDL3 target genes involved in shoot regeneration. **a** Venn diagram of the genes with hyper H3K4me2 in *ldl3-1* compared with WT (C14), the genes upregulated upon shoot induction in WT ($p < 0.01$, FC > 1.25) (S1), the genes down-regulated in *ldl3-1* compared with WT upon shoot induction ($p < 0.01$, FC < 0.8) (UGs_S1), and LDL3-bound genes (C14). **b** Heat maps for gene expression levels (*z*-score of RPM) of putative LDL3 targets (16 genes) in WT and *ldl3-1* before and after shoot induction. **c, d** Shoot regeneration phenotype in the three mutants for three putative LDL3 target genes. Values are mean ± s.d. **$p < 0.01$ (Student's *t*-test). $71 < n < 80$. Scale bars: 5 mm. **e** Histone H3K4 methylation and LDL3 binding patterns in putative target genes. Boxes, exons; thin lines, introns; bold lines at both ends, 5′ or 3′ UTR. See also Supplementary Fig. 7, Supplementary Data 5 and 6. Source data are provided as a Source Data file

spatial patterns of gene expression[20,22], our results indicate an underlying epigenetic regulatory layer that does not directly influence transcriptional output of callus; rather, *LDL3* is up-regulated and presumably removes H3K4me2 during callus formation, which then may allow the genes for shoot initiation to be expressed after shoot-inducing treatments (Fig. 8). Therefore, it is possible that the shoot regenerative competency of callus tissue is acquired through the reduction of H3K4me2 mediated by LDL3 demethylase activity. In animals, H3K4me2 is often linked with transcriptional activation[44,47], although its repressive role in gene

expression has been also reported[51]. Our data in *Arabidopsis* shoot regeneration indicates that H3K4me2 plays a role in transcriptional repression rather than activation, possibly in cooperation with other types of H3K4me. These findings raise two questions: first, how LDL3 selects target genes to regulate during callus formation, and second, how reduced H3K4me2 together with other H3K4me marks facilitates gene activation in cytokinin-rich SIM but not in auxin-rich CIM.

For the first question, it has been revealed that key determinants of the target selection of histone demethylases are either

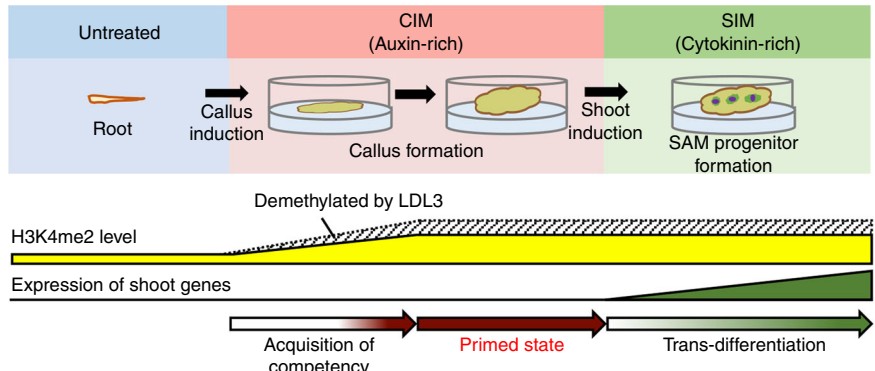

**Fig. 8** Schematic of the acquisition of shoot regenerative competency. LDL3 is up-regulated and erases H3K4me2 marks of downstream genes during callus formation, although the total amount of H3K4me2 is still increased. The removal of H3K4me2 mediated by LDL3 does not immediately alter the gene expression of callus tissue, but is primed for subsequent shoot induction, which allows the genes for shoot initiation to be activated. The shaded area indicates the H3K4me2 removed by LDL3 during callus formation

their own non-catalytic domains (reader domains) or their interacting partners (reader proteins), which read and bind histone modifications[52]. To date, HsLSD1 is found to both directly and indirectly interact with histones, and in the latter case, it is found in several different protein complexes acting on different target genes in different cellular contexts[31,52]. In our study in *Arabidopsis* regeneration, LDL3 is most frequently found on genes with both H3K4me1 and H3K4me3 marks. Thus, it is hypothesized that LDL3 protein is preferentially recruited to the genes with those marks during callus formation. If this is the case, it will be interesting to investigate how LDL3 or its interacting partners recognize H3K4me1 and H3K4me3 marks in callus, and whether this machinery is shared or different between callus and other meristematic tissues where LDL3 is expressed.

For the second question, one possible scenario is that step-wise histone modifications take place between the LDL3-mediated primed H3K4me2 demethylation in CIM treatment and the gene activation in the subsequent SIM treatment. Histone acetylation could be one such modification based on the reports on gene priming for time-lagged expression and LSD1-mediated histone demethylation in other systems. In budding yeast, it has been reported that pre-acetylation mediated by GCN5 histone acetyltransferase (HAT) at silenced promoters allows rapid activation of genes once the mating type of the cells is switched[53]. In addition, in human gastrointestinal endocrine cells, HsLSD1-mediated H3K9me2 histone demethylation is found to facilitate subsequent histone H3K9 acetylation catalyzed by HAT, leading to gene activation[54]. In this case, the protein complex containing HsLSD1 co-occupies gene promoters with basic helix-loop-helix (bHLH) transcription factor NeuroD1, which also associates with HAT. Likewise in *Arabidopsis* shoot regeneration, HAT might play roles in LDL3-mediated gene priming. In the light of these models in animal cells, future investigations should test possible interactions and co-occupations of HAT and LDL3 on target genes primed in the callus and activated during shoot regeneration. A second, non-exclusive scenario to be tested involves the recruitment of specific transcription factors onto primed genes, similarly to the case of pluripotency transcription factor-based gene priming observed in mammal ES cells. In either case, the delayed transcriptional activation in SIM would be explained. Given that the alteration of gene expression in the *ldl3* mutant was observed already at the initial stage of SIM treatment (S1), a key molecule for the regulatory system might be in SIM. This could be cytokinin, which is present at higher levels in SIM than in CIM. Whether phytohormone-responsive transcription factors are involved in the activation of primed target genes remains to

be determined. Since we found that the effect of H3K4me2 on gene expression might be associated with the status of other H3K4me marks, H3K4me marks may play a role in concert in recruiting the key molecule or in releasing an inhibitor of the key molecule. To determine what molecule works downstream of LDL3 to activate specific genes upon shoot induction will give insight into the mechanism by which primed epigenetic states shape the transcriptional landscape in a time-delayed developmental process in plant regeneration.

Three genes were identified as possible LDL3 downstream target genes involved in shoot regeneration. *CIPK23* is implicated in cellular ion homeostasis including that of ammonium ions, and *GLT1* is involved in ammonium assimilation to produce glutamate[55–58]. Given that ammonium influences the shoot regeneration rate in other plant species[59,60], it is possible that these genes regulate shoot regeneration via ammonium signals or metabolic pathways. In this case, during the initial phase of shoot induction, the explants might establish the appropriate circumstances, such as ion balance, for the shoot progenitor cells to form and develop. The role of *UPL4* has not been described in *Arabidopsis*, but ubiquitination activity is strongly suggested based on its sequence[61], it could degrade a protein that inhibits shoot regeneration. It is expected that characterization of these genes will provide clues about the molecular mechanism that establishes the platform for initial shoot fate specification in de novo shoot regeneration.

*ldl3* plants do not show drastic phenotypes in spite of *LDL3* expression in meristems. This could be because of functional redundancy of LDL family genes, compensation by other factors regulating the homeostasis of H3K4me2, or the primed states mediated by *LDL3*, which create domains of competence that are dormant until the environment stimulates regeneration. In normal development, the microenvironment of the plant meristem is perhaps not exposed to such drastic changes in environmental stimuli, as root and shoot cell lineages are separated at the first cell division of the fertilized egg and cells do not generally change their fate beyond their lineage. In contrast, in in vitro shoot regeneration, the alteration of auxin/cytokinin ratios in the explant media causes fate conversion from root-like to shoot. Although *LDL3* expression and its encoded demethylase activity in meristems do not immediately impact on the transcriptional landscape, this machinery may prepare meristems for extreme changes of environmental stimuli. It is tempting to speculate that H3K4 methylation is an epigenetic barrier to plant regeneration, which is overcome specifically in meristems and callus. If this is the case, LDL3-mediated H3K4me2 removal would be one of the

mechanisms that underlie the remarkable developmental plasticity of plant cells in response to environmental alterations; and the degree of developmental plasticity might be controlled by pre-existing histone methylation states, which prime genes in advance of signal inputs. In animals, the knockout of pluripotency-associated genes, including *LSD1*, causes embryonic lethality[62,63], indicating the common use of those genes in the critical processes of acquisition of pluripotency and embryonic development. In plants, on the other hand, the molecular mechanisms for acquisition of pluripotency appear to be different from the ones for normal development, which may enable plant cells to keep their unique regenerative competency in non-embryonic tissues.

## Methods

**Plant materials and growth conditions**. The mutant alleles used in this study were: *lsd1* (*swp1-1*) (SALK_142477), *ldl2-2* (SALK_135831), *fld-3* (SALK_075401), *ldl3-1* (GABI_092C03), *ldl3-2* (SALK_146733), *cipk23* (SALK_032341), *upl4* (SALK_091246), and *glt1* (SALK_079698)[32,33,55,64], all of which are on a Columbia (Col-0) background. The generation of marker lines is described below. Multiple mutants and combinations of the *ldl3-1* mutant and markers were made by crossing. Plants homozygous for both mutants and markers were selected by genotyping and antibiotics. Plants were grown on soil or MGRL medium[65] under long day (16 h light/8 h darkness) photoperiods.

**Regeneration assays**. Root explants (0–1 cm from the root tip) were excised from seedlings six days after germination, and cultured on CIM containing Gamborg's B-5 medium (Wako) with 20 g/l glucose (Wako), 0.5 g/l MES (Wako), 1·Gamborg's vitamin solution (Sigma), 500 µg/l of 2,4-D (Sigma), 50 µg/l of kinetin (Sigma), and 0.8% Gellan gum (Wako), with the pH adjusted to 5.7 using 1.0 M KOH. Continuous light was used for standard callus induction.

After culturing for 14 days on CIM, the explants were transferred onto SIM containing Gamborg's B-5 medium, 10 g/l sucrose, 0.5 g/l MES, 1·Gamborg's vitamin solution, 2 µg/ml trans-zeatin, 0.4 µg/ml indole-3-butyric acid, 1 µg/ml d-biotin, and 0.8% Gellan gum, with the pH adjusted to 5.7 using 1.0 M KOH. Continuous light was used for shoot induction.

After culturing for 12 days on SIM, the number of shoots produced on each explant was evaluated, counting a visible apical meristem surrounded by 2–3 leaves with trichomes as one shoot. All phenotypic assay and microscopic observation experiments were replicated at least three times.

**Microscopic imaging**. For the observation of GFP and CFP (mTurquoise2) fluorescent marker lines, 10 mg/ml of propidium iodide (PI) (Sigma) was applied to samples prior to imaging for the counterstaining of cell outlines. For RFP (dsRed) marker lines, the autofluorescence signal was captured to visualize the tissue outlines. Root explants at the early stages of callus induction (days 0–7 on CIM) were observed using an Olympus FV1200 confocal microscope with a UPLSAPO20X (N. A. = 0.75, W.D. = 0.6 mm) objective lens (Olympus). To detect PI staining, a 559 nm laser line was used for excitation and a 575–675 nm band pass filter was used to collect the signal. To detect the GFP signal, a 473 nm laser line was used for excitation and a 490–540 nm band pass filter was used for signal collection.

Root explants at the late stages of callus induction and at all stages of shoot induction (day 14 on CIM and days 1–7 on SIM), and the SAM in the young seedlings (three days after sowing) and the reproductive stage of plants (after bolting) were observed using an Olympus FVMPE-RS multiphoton microscope with a XLPLN 25XWMP2 (N.A. = 1.05, WD = 2.00 mm) water immersion objective lens (Olympus). For the detection of CFP and PI signals, the laser setting was tuned to 820–850 nm (for CFP) with fixed 1040 nm (for PI). Light was first separated by an FV30-SDM570 dichroic mirror. The CFP signal was collected using an FV30-FCY filter mounted in front of gallium arsenide phosphide (GaAsP) photomultiplier tubes (PMTs). The PI signal was collected using an FV30-FGR filter mounted in front of multi-alkaline PMTs. To detect GFP and PI signals, the laser setting was tuned to 920 nm (for GFP) with fixed 1040 nm (for PI). All light was reflected by an FV30-SDM-M mirror. Signals were collected using an FV30-FGR filter mounted in front of GaAsP-PMTs. To detect autofluorescence and RFP (dsRed) signals, the laser setting was tuned to 920 nm (for autofluorescence) with fixed 1040 nm (for RFP). Excitation laser was transmitted through FV30-NDM760 dichroic mirror. Emitted light was first reflected by an FV30-NDM760 dichroic mirror, filtered through an FV30-BA750RXD barrier filter, and separated by an FV30-SDM570 dichroic mirror. Autofluorescence signal was collected using an FV30-FVG filter mounted in front of GaAsP-PMTs. RFP (dsRed) signal was collected using an FV30-FRCY5 filter mounted in front of multi-alkaline PMTs. The Z-stacks were reconstructed into a projection view using Fiji software (https://fiji.sc/). More than 10 samples were imaged for each marker line to confirm that observed patterns were representative of the respective markers. The region of RAM was defined as the area proximal to QCs where the length/width ratio of cortex cells < 1. The total length of cortex cell files in the RAM region was measured by detecting PI staining using an FV1200 confocal microscope and Fiji software.

The number of nuclei marked by *pWUS::mTq2* in plants (at the stages described above) was counted using Imaris software (Bitplane). Spot signals with 3.5–5.0 µm in diameter in the SAM were extracted and counted after the 3D construction of confocal Z-stacks.

**Generation of transgenic plants**. To make the *pLDL3::LDL3-GFP* fusion construct, an *LDL3* genomic DNA fragment (2.5 kb upstream of ATG to the stop codon) was amplified by PCR with the primers 5′-CACCACTTTTTTCAGG-CAAGCAAAATC-3′ and 5′-AGGTTTTGGAACTTGAGGTATC-3′ and cloned into the pENTR/D-TOPO plasmid (Thermo Fisher Scientific) to create *pENTR_pLDL3::LDL3*. The *LDL3* genomic sequence was recombined into the pMDC107 binary vector upstream of GFP using the Gateway LR Clonase II Enzyme Mix (Thermo Fisher Scientific) to create *pMDC107_pLDL3::LDL3*, which was used for the transformation of plants (wild-type Col-0 for reporter generation and *ldl3-1* for complementation experiments). Transformation was performed by *Agrobacterium*-mediated floral dipping[66], and transgenic plants were selected for hygromycin resistance.

To make *pLDL3::GUS*, the *LDL3* promoter sequence (2.5 kb upstream of ATG) was amplified by PCR with the primers 5′-CACCACTTTTTTCAGGCAAGCAA AATC-3′ and 5′-ACCTATCTACTTATTCCTTAC-3′ and cloned into the pENTR/D-TOPO plasmid to create *pENTR_pLDL3*. The *LDL3* promoter sequence was recombined into the pMDC164 binary vector upstream of GUS to create *pMDC164_pLDL3*, which was used for the transformation of wild-type Col-0 plants.

To make *p35S:LDL3-GFP*, the *LDL3* genomic coding sequence was amplified by PCR with the primers 5′-CACCATGGATGGTAAGGAGAAGAAA-3′ and 5′-AGGTTTTGGAACTTGAGGTATC-3′ and cloned into the pENTR/D-TOPO plasmid to create *pENTR_LDL3*.

To make *p35S:mLDL3-GFP*, the LDL3K949A genomic coding sequence, which contains a point mutation (lysine 949 to alanine) in the amino oxidase domain[28,67], was PCR-amplified from *pENTR LDL3* with primers 5′- GTTTTGGAGTTCTA AATGCAGTTGTTTTGGAGTTCCCAAC-3′ and 5′- GTTGGGAACTCCAA AACAACTGCATTTAGAACTCCAAAAC-3′ to create *pENTR_mLDL3*.

LDL3 and mLDL3 coding sequences were recombined into the pGWB505 binary vector downstream of the 35S promoter and upstream of GFP to create *pGWB505_LDL3* and *pGWB505_mLDL3*, which were used for the transformation of *Nicotiana benthamiana*.

The *pWUS::mTq2* reporter constructed as follows. The *WUS* promoter consists of 4.4 kb of upstream sequence and 1.5Kb of downstream sequence. These fragments were cloned into the pMOA binary vector[60] and a gateway recombination cassette was cloned as a Sma1 fragment between the *WUS* promoter and 3′UTR 1.5 kb fragment. A 2x fusion version of mTurquoise2 with the N7 localization tag was introduced into the plasmid by standard LR recombination with LR clonase II. Transgenic plants were selected for Basta.

**In vivo demethylation assay**. Half of each tobacco leaf was infiltrated with *Agrobacterium tumefaciens* GV3101 strains containing *p35S:LDL3-GFP* or *p35S: mLDL3-GFP* to express *LDL3-GFP* or *mLDL3-GFP*. The other half of the leaves were used as controls. Four days after infiltration, the tobacco leaves were cut into 0.5 cm × 0.5 cm pieces and fixed in 4% paraformaldehyde in phosphate-buffered saline (PBS; 137 mM NaCl, 8.1 mM NaH$_2$PO$_4$, 2.68 mM KCl, 1.47 mM KH$_2$PO$_4$) for 30 min. Nuclei were isolated by chopping leaves in NIB (10 mM Tris HCl, pH 9.5 10 mM EDTA, 100 mM KCl, 0.5 M sucrose, 4 mM spermidine, 1.0 mM spermine, 0.1% (vol/vol) 2-mercaptoethanol). The homogenate was filtered through 30 µm mesh, centrifuged at 600× *g*, and resuspended in 50 µL PBS. The suspension was then dropped onto a clean slide and dried at 4 °C, incubated in 0.2% Triton X-100 in PBS for 10 min at room temperature, and washed with PBS. Slides were incubated by 4% BSA in PBS at room temperature for 30 min and subjected to immunostaining. The antibodies used for labeling GFP and histone methylation were anti-GFP mouse (Merk 11814460001) 1:100; H3K4me1 (Abcam ab8895) 1:100; H3K4me2 (Abcam ab32356), 1:100; H3K4me3 (Abcam ab8580), 1:100. The histone antibodies were revealed by Alexa Flour 546-conjugated goat anti-rabbit (Thermo Fisher Scientific A11035, 1:1000) and the GFP was revealed by Alexa Flour 488-conjugated goat anti-mouse (Thermo Fisher Scientific A11001, 1:1000). After immunostaining, the DNA was stained with 0.1 µg/mL DAPI (Roche Diagnostics GmbH, Penzberg, Germany) and the slides were mounted with VECTASHIELD mounting medium (Vector Laboratory). The samples were observed by fluorescence microscope (BX53, Olympus, Tokyo, Japan) with a ×40 objective (UPLSAPO40X, Olympus, Tokyo, Japan) and a CCD camera (DOC CAMU3-50S5M-C, Molecular Devices) controlled with MetaVue (Molecular Devices). The images were analyzed using ImageJ (https://imagej.nih. gov/ij/). The transfected nuclei with GFP signal versus non-transfected nuclei without GFP signal (control) were observed. A maximum of five control nuclei per one transfected nucleus were randomly picked up from the same field and assessed for signal intensity. Fewer than five non-transfected nuclei without GFP signal (control) per one transfected nucleus with GFP signal were randomly picked up from the same field and assessed for signal intensity.

**GUS staining**. Seedlings (6 days after germination) and calli (14 days on CIM) from *pLDL3::GUS* lines were treated with 5 mM of ferricyanide and ferrocyanide. Samples were observed with a stereomicroscope equipped with a DP72 digital camera (Olympus).

**RNA-seq**. Root explants derived from wild type (Col-0) and *ldl3-1* seedlings were collected on days 0 and 14 on CIM (C0, C14), and days 1 and 7 on SIM (C14S1, C14S7). Total RNA was isolated from the collected explants using the PureLink Plant RNA Reagent (Thermo Fisher Scientific). The integrity of purified RNA was assessed using a 2100 Bioanalyzer (Agilent). A total of 1000 ng RNA was used to construct a transcriptome library with TruSeq RNA Sample Preparation v.2 (Illumina). Libraries were pooled and 36–86 bp single-read sequences were obtained with a NextSeq 500 sequencer (Illumina). Three independent biological replicates were analyzed for each genotype.

**RNA-seq data analysis**. Quality-filtered reads were mapped onto cDNA sequences of annotated genes and other transcripts of TAIR10 using Bowtie[68] with --all --best --strata settings (Supplementary Data 1 and 2). Differentially expressed genes (DEGs) were identified in R using the R package edgeR ver 3. 16. 5[69], treating biological triplicates as paired samples. Genes with adjusted $p$ values < 0.01 and FC > 1.25 or < 0.8 in each comparison were identified as DEGs (list of datasets are provided in Supplementary Data 3). Comparison of the relative expression levels of DEGs between conditions (at three different stages in WT and *ldl3*) (Fig. 3c–f) was performed as follows: half of the minimum RPKM value was added to all RPKM values, and the average value of three biological replicates for each gene and each condition was normalized to that of C14 in WT. To calculate the relative gene expression (*ldl3*/WT) (Figs. 5c, f, 6b, Supplementary Figures 5, 6), half of the minimum RPKM value was first added to all RPKM values, prior to all other procedures for constructing box plots.

**ChIP-seq**. Root explants derived from wild type (Col-0) and ldl3-1 seedlings were collected on day 0 and 14 on CIM (C0, C14), and day 1 on SIM (C14S1); 0.1 g of explants was frozen with liquid nitrogen, ground into fine powder with SH-48 (Kurabo), cross-linked, and nuclear-extracted in the nuclei isolation buffer (1% formaldehyde, 0.6% TritonX-100, 14.4 mM 2-mercaptoethanol,) with 1 mM Pefa-bloc SC (Merk) and complete protease inhibitor cocktail (Merk). Sonication was conducted using S2 or M220 focused ultrasonicators (Covaris) and milliTUBE 1 ml AFA Fiber (Covaris). Sonicated samples were incubated with the antibody at 4 °C for overnight. The antibodies used were: rabbit anti-H3K4me1 (ab8895; Abcam), rabbit anti-H3K4me2 (ab32356; Abcam), rabbit anti-H3K4me3 (ab8580; Abcam), and rabbit anti-H3 (ab1791; Abcam). Protein G Magnetic Dynabeads (Thermo-Fisher Scientific) were used for immuniprecipitation. The beads were washed with low-salt RIPA buffer [50 mM Tris·Hcl, pH 7.8, 150 mM NaCl, 1 mM EDTA, 1% Triton X-100, 0.1% SDS, 0.1% Sodium deoxycholate and 1% Complete protease inhibitor (Roche)], twice with high-salt RIPA buffer [50 mM Tris·Hcl, pH 7.8, 500 mM NaCl, 1 mM EDTA, 1% Triton X-100, 0.1% SDS, 0.1% Sodium deoxycholate and 1% Complete protease inhibitor (Roche)], with LNDET buffer (250 mM LiCl, 1% IGEPAL, 1% Sodium deoxycholate, 1 mM EDTA, 10 mM Tris-HCl pH 7.8) and then with TE buffer. After the elution buffer (10 mM Tris-HCl pH 7.8, 0.3 M NaCl, 5 mM EDTA, 0.5% SDS) was added to the beads, the beads were incubated overnight at 65 °C. The lysis was treated with 200 ng/ml RNaseA at 37 °C for 30 min and then treated with 800 ng/ml Proteinase K and 400 ng/ml glycogen at 37 °C for 2 h. After phenol chloroform extraction and ethanol precipitation, the pellet was suspended in Buffer EB (Qiagen). Collected DNA was quantified with the Qubit dsDNA High Sensitivity Assay kit (Thermo Fisher Scientific), and 1 ng DNA was used to make a library for Illumina sequencing. The library was constructed with the KAPA Hyper Prep Kit for Illumina (KAPA Biosystems), and dual size selection was performed using Agencourt AMPure XP (Beckman Coulter) to enrich 300–500 bp fragments. Libraries were pooled, and 75 bp single-read sequences were obtained with the NextSeq 500 sequencer (Illumina).

Genome-wide localization patterns of the LDL3 protein were analyzed using *pLDL3::LDL3-GFP* transgenic plants in a *ldl3-1* background (*pLDL3::LDL3-GFP/ldl3-1*) and control transgenic plants expressing *p35S::GFP* in a wild type (Col-0) background (*p35S::GFP*). Calli at 14 days on CIM (C14) derived from *pLDL3::LDL3-GFP/ldl3-1* and *p35S::GFP* plants underwent ChIP-seq analysis using an anti-GFP rabbit antibody (ab290; Abcam) as described above. Two independent biological replicates were analyzed for each genotype at each stage.

**ChIP-seq data analysis**. Quality-filtered reads were aligned onto the *Arabidopsis* reference genome TAIR10, using Bowtie with -m 1 -S parameters to report only uniquely mapped reads. The resulting SAM files were converted to sorted BAM files using SAMtools[70], then converted to BED files using BEDTools[71]. The "slop" function of BEDTools was used to extend the 5′ end of ChIP-seq reads toward the 3′ direction to fit the average insertion size (250 bp) of the sequenced libraries. Then, the "coverage" function of BEDTools was used to calculate the number of reads that overlapped with each annotation unit (Supplementary Data 1 and 2). LDL3 binding sites were detected using Model-based Analysis for ChIP-seq (MACS2[72]) with reads from the anti-GFP (*p35S::GFP*) sample used as controls ($q < 0.1$) (list of datasets are provided in Supplementary Data 5). For visualization, TDF files were created using igvtools (extension factor: 200) from BAM files, and visualized with Integrative Genome Viewer[73]. The ngs.plot.r program[74] was used to determine the methylation profile and LDL3 binding sites around gene bodies. All downstream analyses including figure plotting and statistical analyses were conducted in R. The scatter plots and NGS plots for ChIP-seq results are shown for one of the biological replicates, because two replicates showed very high reproducibility. To calculate the methylation level ratio (*ldl3*/WT) (Figs. 5c, f, and 6b), half of the minimum RPKM value was added to all RPKM values, and then genes were sorted as indicated in the figure legends.

**Reporting summary**. Further information on experimental design is available in the Nature Research Reporting Summary linked to this article.

## Data availability

ChIP-seq and RNA-seq data associated with this study have been deposited in DDBJ Sequence Read Archive (DRA) under the accession number, DRA008014, and NCBI Sequence Read Archive (SRA) under the accession number, SRP187025. The source data underlying Figs. 4e, 7d, and Supplementary Figure 1a, b, c, 2b, 6b and d are provided as a Source Data file. The authors declare that any other data supporting the findings of this study are available within the manuscript and its supplementary files or are available from the corresponding authors upon request.

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

## Acknowledgements

We thank V. A. Grieneisen for *pPIN1::PIN1-GFP* seeds, the European Arabidopsis Stock Center (NASC) for *ldl3-1* seeds, and the Salk Institute Genomic Analysis Laboratory Arabidopsis Biological Resource Center (ABRC) for *ldl3-2*, *lsd1* (*swp1-1*), *ldl2-2*, *cipk23*, *upl4*, and *glt1* seeds. We also thank Brian Ma and Edward Pursifull at the California Institute of Technology for technical assistance with initiating the project and screening the mutants, Keiko Yoda and Yukiko Kawaguchi at Tokyo University of Science for technical assistance, Adrienne Roeder for critical reading and comments on the manu-script, and Sarah Williams, PhD, from Edanz Group (www.edanzediting.com) for editing a draft of this manuscript. This research was supported by CREST grants from the Japan Science and Technology Agency (JPMJCR13B4) and MXT/JSPS KAKENHI (15H05955 and 15H05962) to S.M., MXT/JSPS KAKENHI (15H05963) to T.K. and by the Howard Hughes Medical Institute to E.M.M.

## Author contributions

K.S., H.I., and S.M.: Conception, design and interpretation of data; K.S. and E.M.M.: Project initiation and mutant screening; H.I.: Final mutant screening; H.I., H.T., and S.K.: Acquisition of data; K.S., H.I., H.T., and S.K.: Analysis and interpretation of data and drafting the article; K.S., P.T., E.M.M. and S.M.: Revising the article; P.T., Y.I., T.Sasaki.,

M.S., E.M.M. and S.M.: Generation of unpublished materials; S.K., Y.I., and H.T.: Observation of reporter lines; T.Sakamoto and M.A.: Technical support for microscopic observation, construction and western blotting analysis; S.I., H.T., M.K., and T.K.: Technical support for RNA-seq and ChiP-seq analysis; and T. Suzuki.: Sequencing. The manuscript was written based on inputs from all authors.

## Additional information

**Supplementary Inform tion** accompanies this paper at https://doi.org/10.1038/s41467-019-09386-5.

**Competing interests:** The authors declare no competing interests.

