## [Peer Review File · Nature Communications]

Reviewers' comments:

Reviewer #1 (Remarks to the Author):

The manuscript entitled "Primed histone demethylation regulates shoot regenerative competency" by Ishihara and Sugimoto et al reports the role of LDL3-mediated H3K4me2 demethylation in de novo shoot regeneration from callus. Plants have powerful regenerative abilities. Molecular mechanism of shoot formation from callus is an interesting and timely topic in the field of plant regeneration. This manuscript nicely showed a novel mechanism involving the epigenetic control in promotion of shoot-related genes during shooting from callus. Two key findings are: 1) Mutation in LDL3 results in shooting defect from callus; 2) LDL3 is involved in the genome-wide control of H3K4me2 demethylation in callus cells, and this is responsible for upregulation of shoot-related genes on SIM. Overall, I enjoyed reading this story, and I believe that these findings will improve our understanding of how chromatin factors are involved in shoot regeneration in tissue culture. I have several suggestions for the authors to improve the manuscript.

1) It will be much more convincing if the authors use an in vivo histone demethylation assay in tobacco leaves to show that LDL3 is specifically involved in H3K4me2 demethylation (Nat Genet. 2011 Jun 5;43(7):715, Figure 1) with H3K4me1, H3K4me3 and H3K27me3 as controls.

2) The authors nicely showed that CIPK23, UPL4, GLT1 as direct targets of LDL3 in shoot regeneration. How about expression patterns of these genes? Are they shoot-specific genes?

3) Line 98, 'this root identity is crucial for shoot regenerative competency'. A recent study showing that root-primordium-related genes are required for shoot regeneration from callus could be cited (Plant Cell Physiol. 59(4): 739-748)

4) Line 357, The finding of ARR12 regulation by LDL3 is interesting, but I am not sure that 'moderately (but not significantly)' is a proper description. I would like to suggest the authors to make a more careful description on this result to avoid misleading statement.

Reviewer #2 (Remarks to the Author):

Review assessment of „Primed histone demethylation regulates shoot regenerative competency“

Plants have the ability to regenerate relatively easily, especially in vitro, from tissue fragments or young seedlings cultured in the presence of phytohormones. Taking advantage of a robust in vitro system in the model plant Arabidopsis, and a long-lasting expertise in the field, the authors report on a novel epigenetic regulator necessary for the acquisition of regeneration competence. The study builds on previous work from the authors where transcriptome profiling allowed identifying candidate genes markedly associated with the step-wise regeneration process. Using a forward genetic approach they identified a putative histone H3 Lysine 4 demethylase, LDL3, necessary to reprogram the H3K4me2 landscape in callus cells and prime them for a regenerative program. The work is elegant, well substantiated with thorough transcriptional and epigenetic profiling data, reporter gene and phenotypic analyses. The manuscript is well structured and generally well presented but I have some suggestions for minor (in content, but the number of items is long) text revision aiming at improving the clarity and conciseness of the manuscript. The discussion should perhaps also summarize a bit more the results and be less speculative or more concise on the models to be tested. In addition, a biochemical approach testing LDL3 enzymatic activity and substrate preference is not presented. Thus, while the data all converge towards the likely model that LDL3 is a K4me2 demethylase, this little caveat - not affecting the main conclusion- could be discussed. But importantly, the findings are novel, timely and with broad implications for the field

of research aiming at elucidating pluripotency mechanisms in plants and animals.

P3-4. Introduction . The first sentence "acquisition of pluripotency by somatic cells in multicellular organisms is achieved by the removal of epigenetic memories" suggest that this is a general concept applied to plants and animals. Although it is likely to hold true also in plants, we still do not have a clear view of epigenetic reprogramming during regeneration (addressed by this paper). In addition the following paragraph deals mostly with our understanding of pluripotency in animal systems. Please be specific for the reader whenever concepts are borrowed from plant or animal research or both.

P5 L74 "forced expression"-> ectopic expression

L84-87 "during callus formation, a step-wise acquisition of competency for events involved in organ regeneration...for complete organ regeneration" long sentence and not clear, please reformulate -> suggestion for "Regeneration competence is thought to be acquired step-wise with a period of callus growth being a prerequisite to organ formation" ?

P11-12: Please revise the description of comparative transcriptional profiles . This is currently hard to follow and does not project a clear message. Maybe avoid "the downregulated genes were eventually downregulated in Idl3 as well as in wild type although the decline of expression of early down regulated genes in Idl3 was small compared with wildtype at the initial stage.." (L182-L184 but many examples through the pages 11-12). Choose a different concept of presentation, perhaps using the idea of kinetics or rate of expression (+ for up and – for down regulation, over time) to compare. The heatmaps do not allow to visualize easily the downregulated genes that are less downregulated etc. Why not using a scatter plot of the fold changes over time, wt (x) against mutant (y)? The upper right quadrant would show genes upregulated during culture and deviation from the median shows the genes with faster or slower kinetics in the mutant; same with lower left quadrant (downregulated genes etc.)

Figure 3: Graphs in c-f are a choice of representation which do not convey very well the message. The box plots show a high variation in expression levels among the classes, but small trends looking at the median value among large groups of genes - for instance Idl3 DGs show a moderate average fold change with a mean ratio of 0.5 but the variation is large. Hence it may be more informative to have a scatter plot , or better, a volcano plot representation with colors highlighting the differentially expressed genes for a given P value and Fold change, and a another plot showing the shift in expression of these SIM –regulated genes , in Idl3 . Alternatively, or in addition, it may be usefu to adopt a different concept, talking about kinetics (or rate) of gene activation/repression upon culture, which seem affected in Idl3. see also comment for Figure 5

L217 "enhanced H3K4me2 and H3K4m3 in wild-type explants": replace for "increased...levels"

L218-219 "in Idl3 explants, H3K4me2 increased compared with wild-type": it is difficult to understand because the levels also increases in wt during culture. Did you mean, "in Idl3 explants, H3K4me2 levels were more dramatically increased than in wild-type" or "..also increased but with a higher amplitude/fold-change"?

L233-234 "LDL3 binds to the gene body and correlates there with the removal of H3K4me2 although the total amount of H3K4me2 is increased during callus formation". Confusing, how do you explain this apparent contradiction?

L240 another shortcut "the enhancement of H3K4me2 during callus formation" : change for "increased levels of H3K4me2 during.."

L244 "H3K4me2 is maintained beyond shoot induction without additional removal" : difficult to talk about maintenance here, so suggestion for "LD3 does not influence the levels of H3K4me2 during

shoot induction"

L255 "biphasic curve of correlation (figure S5)" is not at all obvious – I see a (linear?) correlation for categories i-iii featuring low to intermediate gene expression classes. High expression level classes do not seem represented in H3K4me2 profiles, instead, intermediate expressed genes are saturated with increasing levels of H3K4me2 : are they the same that are correlated with increasing H3K4me3 amount? In other words, does the level of K4me2 predict well K4me3 levels?

L261 "genes with a high degree of H3K4me2 modification (ldl3/WT) in calli" it is difficult to understand whether modification relates to K4me2 or to the changes induced by the mutation -> "changes in H3K4me2 levels in ldl3 calli did not correlate with gene expression changes at the same stage but instead with changes during shoot induction, indicating that H3K4me2 levels in calli primed for expression states after induction". In addition, is it possible to interpret that the negative correlation between K4me2/expression changes suggests an inhibitory effect of H3K4me2 on transcription?

Figure 5 could be simplified:

Figure 5a: remove the panels with red/blue pseudo coloring –do not bring a key message
Figure 5c. the message is not clearly conveyed. Currently the mean gene expression ratio of ldl3/wt are more or less around one, thus not highlighting the big changes discussed in the text. Reading carefully the figure , one can understand that the ratio classes i-iv correspond to highly variable gene expression but expression does not correlate with the fold difference of K4me2 between ldl3 and wt; ratio classes v-viii correspond to more robustly expressed genes and their expression is not correlated with the K4me2 fold change. ? A scatter plot would be more informative and easier to read, plotting the fold changes in K4me2 against the fold changes in expression, with colors to indicate the groups of genes significantly deviating from the median or colors to differentiate C14 and C14S1 profiles. Eventually (instead or in addition), the authors should consider showing the K4me2 levels only for the genes with changes in expression levels in the WT and genes where the kinetics is altered in the mutant (hence only a subset of the former). Rest in Sup data.

Figure 5d. what is the y axis? Expression or K4me2? Again here, why not showing a scatter plot, more intuitive than the classes low-high K4me2 ratios ldl3/wt

Figure 5e-f could go in supplemental data

Figure 6 should be simplified

Suggestion for 6a: Venn diagram of me1-, me3, me2-marked loci and LDL3 (or only me1, me3 and LDL3 but need to justify this approach, currently not clear) eventually together with a, complementary PCA analysis with me1, me2, me3, LDL3 profiles together with ldl3 misregulated genes – this should show the components that better describe LDL3 targets

Suggestion for 6b: call clearly the plots "LDL3 targets associated with": "H3K4me3 and H3K4me1", "H3K4me1 only", "H3K4me3 only", "no H3K4 methylation" –

but here again, I would suggest a scatter plot, or a different representation. Currently all the mean ratio of gene expression ldl3/WT are close to 1 thus does not highlight the misregulated genes.

If the authors want to keep box plots, they should consider to separate the LDL3 targets that are misregulated and analyse their preferential H3K4me1, me2 and me3 levels in WT and mutant.

Choose one clear message and place the rest in the sup data

L290-291 and later + figure 6: the choice to add a "+" or a "-" to me1, me2, me3 to describe their absence/presence is not conventional and creates a complex setup for the text and the figure 6.

L271-280. The term bivalent modification is valid only for the case where H3K4me and H3K27me are found on the *same* nucleosome. This can be addressed only in sequential ChIP experiments, but not in the experimental set up done here

L289-L322 This paragraph is confusing. I suggest the authors to choose a different concept of presentation, perhaps starting with a PCA analysis (me1, me2, me3, LDL3 target, ldl3 misregulated genes) to show which marks describe best the LDL3 targets, or if subgroups of genes are visible. The Venn diagram Figure 7a is good and comes as a complement to the PCA.

L325-L335 I would drastically simplify the paragraph in 1-2 sentences (To identify LDL3 target genes involved in shoot regeneration, we selected 16 genes bound by LDL3, upregulated in wildtype upon shoot induction but downregulated in ldl3 and hypermethylated on H3K4(me2) at the callus stage (Figure 7a).

L345-354 belongs to discussion

L375-376 "shows that H3K4me2 plays a role in transcriptional repression rather than activation" is a not well substantiated. The data do not show clearly a causality. Are high levels of H3K4me2 associated with low gene expression levels in wild-type? (the data focus on the fold changes in the mutant and wild type), do K4me2 levels increase on SIM as they do on CIM between C0 and C14 (Supp fig4) ? In fact The Supp fig5 show that high levels of K4me2 do not correlate with higher gene expression levels while high K4me3 levels do. But there is no indication that K4me2 represses gene expression. Instead, the fact that LDL3 target genes show lower expression in the mutant together with higher K4me2 suggests that the removal of this mark OR the stepwise methylation towards me3 state may be necessary to sustain high levels of transcription . How do the levels of K4me3 compare for these specific ldl3 downregulated genes?

L393-395 Please clarify the idea of stepwise histone methylation / DNA methylation- why DNA methylation?

L405-406 " HAT might play roles" , "it might be possible that".. it reads like overspeculation. Instead the authors could mention that in the light of this model in animal cells, future investigation should test possible interaction and co-occupation of HAT and LDL3 on target genes primed in callus and activated during shoot regeneration

L409 "or more directly, another possible scenario" ♦ " a second, non exclusive scenario to be tested involves the recruitment of specific TF by LDL3 onto primed genes"

L413-421: please reformulate for more concise and less speculative and/or describe a model to be tested ("however under this hypothesis [above], the delayed transcriptional activation remains to be explained. Whether phytohormone-responsive TF primed on target genes are involved remained to be determined"

L425 why the answers to the above questions (which answers?) explain the absence of drastic phenotype in ldl3 plants? What about functional redundancy, compensation by other factors regulating the homeostasy of H3K4me2? Alternatively, LDL3 expression just creates domains of competence which are dormant until the environment stimulates regeneration?

Reviewer #3 (Remarks to the Author):

It is clear that LDL3 has an important function in the participation of shoot generation. However, the conclusions made by the authors are not quite correlated with the results presented. The authors claimed that LDL3 eliminates H3K4me2, however, the results from Suppl. Fig. 4A confirms that there is actually a very high increase in H3K4me1 and H3K4me3, which could be the responsible for such effect. By analyzing all the data presented (Suppl. Data 4B, Suppl. Data 5,

Suppl. Data 6) all indicates that the conclusions that LDL3 works as a histone demethylation are not correct. How the authors can explain the abundant accumulation of H3K4me1 and me3 after callus induction in Suppl. Figure 4B. It is not clear to me why the authors only show three genes (Fig. 7E) to conclude their findings

Furthermore, It is not clear how the primed state is demonstrated in the results.

Specific comments:

In general, the references in the introduction section are quite old.

Page 5 lines 82-84: The authors stated that "in many plant species and tissue types de novo organogenesis is only achieved from pre-induced callus, and not directly from native tissue" However, the cited reference does not state that information. There is plenty of published information about the Novo callus formation (Book Plant Cell Culture Protocols, Springer 2018 <https://link.springer.com/book/10.1007/978-1-4939-8594-4>). Also, the authors need to consider the habituation effect in some cultures in order to make a better discussion.

Page 6 line 103: The authors stated that "Few reports have described the epigenetic basis of the acquisition of shoot regenerative.." however, CIM and SIM studies have been investigated from an epigenetic point of view. (Stroud H, Ding B, Simon SA, Feng S, Bellizzi M, Pellegrini M, Wang G-L, Meyers BC & Jacobsen SE (2013) Plants regenerated from tissue culture contain stable epigenome changes in rice. *eLife* 2 doi:10.7554/eLife.00354. Vining K, Pomraning K, Wilhelm L, Ma C, Pellegrini M, Di Y, Mockler T, Freitag M & Strauss S (2013) Methylome reorganization during in vitro dedifferentiation and regeneration of *Populus trichocarpa*. *BMC Plant Biology* 13(1):92) Figure 8 seems quite deterministic because it shows as a unique player for the acquisition of shoot regenerative competency to LDL3 and the effect of demethylation in H3K4me2. However, as I said, with the presented results, it could be due to the hypermethylation in H3K4me1 and me3. I think that more experiments need to be done in order to confirm that 1) the acquisition of shoot competency is due to H3K4me2 and not by H3K4me1 or me3; 2) that LDL3 is indeed a demethylating gene and not a hypermethylating player; 3) that there is a priming mechanism behind the process.

Point-by-point response to reviewer's comments

Revised portions are highlighted in our revised manuscript. The following pages explain how we have addressed the concerns of the reviewers.

Reviewer #1

Comment 1

1) It will be much more convincing if the authors use an in vivo histone demethylation assay in tobacco leaves to show that LDL3 is specifically involved in H3K4me2 demethylation (Nat Genet. 2011 Jun 5;43(7):715, Figure 1) with H3K4me1, H3K4me3 and H3K27me3 as controls.

Response

We undertook the experiment suggested by the reviewer, and present the results in Fig. 4 and Supplemental Fig. 6. We successfully demonstrated the demethylase activity of LDL3, which acts dramatically on H3K4me2 and slightly on H3K4me3 in the tobacco-based transient expression system. Taken together with the CHIP-seq data in *Arabidopsis* explants, we conclude that LDL3 preferentially removes H3K4me2 during callus formation.

Comment 2

2) The authors nicely showed that CIPK23, UPL4, GLT1 as direct targets of LDL3 in shoot regeneration. How about expression patterns of these genes? Are they shoot-specific genes?

Response

We searched for expression domains of these genes in normal development using the eFP Browser (<http://bar.utoronto.ca/efp/cgi-bin/efpWeb.cgi>) and a previous report (Cheong *et al.*, 2007 Plant J). The genes were found to be expressed in various tissues such as roots, hypocotyls, leaves, and embryos. Thus, we speculate that they are not specifically expressed in shoot progenitor cells during regeneration but are involved in establishment of the tissue environment to facilitate *de novo* shoot regeneration at the early stage of SIM treatment.

Comment 3

3) Line 98, this root identity is crucial for shoot regenerative competency. A recent study showing that root-primordium-related genes are required for shoot regeneration from callus could be cited (Plant Cell Physiol. 59(4): 739-748)

Response

We have cited more papers in the revised manuscript, including the suggested article (line 85).

Comment 4

4) Line 357, The finding of ARR12 regulation by LDL3 is interesting, but I am not sure that “moderately (but not significantly)” is a proper description. I would like to suggest the authors to make a more careful description on this result to avoid misleading statement.

Response

We rephrased this sentence as follows in line 362:

Among them, *ARABIDOPSIS RESPONSIVE REGULATOR 12 (ARR12)* was down-regulated in *ldl3* compared with wild-type, which became significant at *S7* (*S7*, $p = 0.00371$, FC = 0.718; *S1*, $p = 0.02446$, FC = 0.823).

Reviewer #2

Comment 1

P3-4. Introduction. The first sentence “acquisition of pluripotency by somatic cells in multicellular organisms is achieved by the removal of epigenetic memories” suggest that this is a general concept applied to plants and animals. Although it is likely to hold true also in plants, we still do not have a clear view of epigenetic reprogramming during regeneration (addressed by this paper). In addition the following paragraph deals mostly with our understanding of pluripotency in animal systems. Please be specific for the reader whenever concepts are borrowed from plant or animal research or both.

Response

The first two sentences in this paragraph describe phenomena that are widely observed in both animals and plants, although the evidence in plants is limited. However, the remainder of this paragraph and the next focus mostly on animal stem cell studies, as the reviewer pointed out. To clarify this, we added “Stem cell studies in animals described...” at the beginning of the third sentence.

Comment 2

P5 L74 “forced expression”-> ectopic expression.

Response

We revised the texts according to the reviewer’s suggestion.

Comment 3

L84-87 “during callus formation, a step-wise acquisition of competency for events

involved in organ regeneration...for complete organ regeneration” long sentence and not clear, please reformulate -> suggestion for “Regeneration competence is thought to be acquired step-wise with a period of callus growth being a prerequisite to organ formation” ?

Response

We revised the texts according to the reviewer’s suggestion.

Comment 4

P11-12: Please revise the description of comparative transcriptional profiles . This is currently hard to follow and does not project a clear message. Maybe avoid “the downregulated genes were eventually downregulated in ldl3 as well as in wild type although the decline of expression of early down regulated genes in ldl3 was small compared with wildtype at the initial stage..” (L182-L184 but many examples through the pages 11-12). Choose a different concept of presentation, perhaps using the idea of kinetics or rate of expression (+ for up and – for down regulation, over time) to compare. The heatmaps do not allow to visualize easily the downregulated genes that are less downregulated etc. Why not using a scatter plot of the fold changes over time, wt (x) against mutant (y)? The upper right quadrant would show genes upregulated during culture and deviation from the median shows the genes with faster or slower kinetics in the mutant; same with lower left quadrant (downregulated genes etc.)

Response

We would like to depict how every gene of interest changes its expression level depending on the genetic background and stage (two genetic backgrounds and three stages). However, a scatter plot using two different stages does not trace the behavior of every gene at all three stages. Additionally, a scatter plot shows the absolute value of changes in gene expression between different conditions, which does not convey the extent to which genes change expression relative to other conditions. Therefore, we think that the heat map is most suitable to present our conclusions and would like to keep the original figures in the manuscript.

Comment 5

Figure 3: Graphs in c-f are a choice of representation which do not convey very well the message. The box plots show a high variation in expression levels among the classes, but small trends looking at the median value among large groups of genes - for instance ldl3 DGs show a moderate average fold change with a mean ratio of 0.5 but the variation is large. Hence it may be more informative to have a scatter plot , or better, a volcano plot representation with colors highlighting the differentially expressed genes for a given P value and Fold change, and a another plot showing the shift in expression of these SIM –regulated genes, in ldl3. Alternatively, or in addition, it may be useful to adopt a different concept, talking about kinetics (or rate) of gene activation/repression upon culture, which seem affected in ldl3.

Response

We revised the box plots using logarithms in the y-axis to more accurately evaluate relative gene expression levels. The variation of each box is now reduced compared with the original. The text and figure organization were also revised for clarity (lines 182–186). We deleted the phrase “although the speed of cancellation is delayed relative to wild type” from the last sentence of this section (line 211). We also moved the box plots to the supplementary file and added the heat maps to Figure 3 instead.

Comment 6

L217 “enhanced H3K4me2 and H3K4m3 in wild-type explants”: replace for “increased...levels”.

Response

We revised the text according to the reviewer’s suggestion.

Comment 7

L218-219 “in *ldl3* explants, H3K4me2 increased compared with wild-type”: it is difficult to understand because the levels also increases in wt during culture. Did you mean, “in *ldl3* explants, H3K4me2 levels were more dramatically increased than in wild-type” or “..also increased but with a higher amplitude/fold-change”?

Response

We meant the latter, and have clarified this as follows: “in *ldl3* explants, H3K4me2 increased even more than in wild-type during callus formation” in line 218

Comment 8

L233-234 “LDL3 binds to the gene body and correlates there with the removal of H3K4me2 although the total amount of H3K4me2 is increased during callus formation”. Confusing, how do you explain this apparent contradiction?

Response

The total amount of H3K4me2 is increased during callus formation, but LDL3 demethylases H3K4me2 on the genes it associates with during this process. In support of this notion, when focusing on LDL3-bound genes, we observed a decrease in H3K4me2 levels at the gene body (Supplemental Fig. 5b). Additionally, Supplemental Fig. 8a shows a specific increase in H3K4me2 levels of LDL3-bound genes in the *ldl3* callus compared with the wt callus. We describe these points in line 278 of the revised manuscript.

Comment 9

L240 another shortcut “the enhancement of H3K4me2 during callus formation” : change for “increased levels of H3K4me2 during..”

Response

We revised the text according to the reviewer's suggestion.

Comment 10

L244 “H3K4me2 is maintained beyond shoot induction without additional removal” : difficult to talk about maintenance here, so suggestion for “LD3 does not influence the levels of H3K4me2 during shoot induction”

Response

As suggested, it is difficult to know if H3K4me2 is maintained from our data. We therefore clarified this with the addition of “apparently” to this sentence.

Comment 11

L255 “biphasic curve of correlation (figure S5)” is not at all obvious – I see a (linear?) correlation for categories i-iii featuring low to intermediate gene expression classes. High expression level classes do not seem represented in H3K4me2 profiles, instead, intermediate expressed genes are saturated with increasing levels of H3K4me2 : are they the same that are correlated with increasing H3K4me3 amount? In other words, does the level of K4me2 predict well K4me3 levels?

Response

We re-analyzed this point in Supplemental Fig. 7.

Within the box plots of this figure we divided all genes according to expression levels but not by methylation levels. Therefore, the members of each class are the same across the different methylation marks (H3 and K4me1, 2, and 3). K4me2 showed the same tendency as H3 and K4me1, while only K4me3 showed a positive correlation with gene expression levels, as previously indicated. K4me3 levels, but not K4me2 levels, were increased in the high gene expression classes. Therefore, there is no correlation between the levels of K4me2 and K4me3.

The scatter plots of Supplemental Fig. 7b showed that H3, K4me1, and K4me2 are not positively correlated with gene expression levels, but that K4me3 is. This indicated that K4me2 levels do not influence gene expression, and we can exclude the possibility of its saturation effects on the intermediately expressed gene class.

Comment 12

L261 “genes with a high degree of H3K4me2 modification (ldl3/WT) in calli” it is difficult to understand whether modification relates to K4me2 or to the changes induced by the mutation -> “changes in H3K4me2 levels in ldl3 calli did not correlate with gene expression changes at the same stage but instead with changes during shoot induction, indicating that H3K4me2 levels in calli primed for expression states after induction”. In addition, is it possible to interpret that the negative correlation between K4me2/expression changes suggests an inhibitory effect of H3K4me2 on transcription?

Response

We are grateful for this suggestion because the revised figure is now easier to understand.

As the reviewer suggested, we think that hyper-H3K4me2 in *ldl3* calli represses gene activation upon shoot induction. Thus, H3K4me2 removal by LDL3 may promote transcription upon shoot induction in wt calli.

Comment 13

Figure 5 could be simplified:

Figure 5a: remove the panels with red/blue pseudo coloring –do not bring a key message

Figure 5c. the message is not clearly conveyed. Currently the mean gene expression ratio of *ldl3*/wt are more or less around one, thus not highlighting the big changes discussed in the text. Reading carefully the figure, one can understand that the ratio classes i-iv correspond to highly variable gene expression but expression does not correlate with the fold difference of K4me2 between *ldl3* and wt; ratio classes v-viii correspond to more robustly expressed genes and their expression is not correlated with the K4me2 fold change. ? A scatter plot would be more informative and easier to read, plotting the fold changes in K4me2 against the fold changes in expression, with colors to indicate the groups of genes significantly deviating from the median or colors to differentiate C14 and C14S1 profiles. Eventually (instead or in addition), the authors should consider showing the K4me2 levels only for the genes with changes in expression levels in the WT and genes where the kinetics is altered in the mutant (hence only a subset of the former). Rest in Sup data.

Figure 5d. what is the y axis? Expression or K4me2? Again here, why not showing a scatter plot, more intuitive than the classes low-high K4me2 ratios *ldl3*/wt

Figure 5e-f could go in supplemental data

Response

We added a scatter plot as suggested by the reviewer. This indicated that the gene expression levels of C14S1 decreased as the methylation levels of C14 increased (green dots), while the gene expression levels of C14 did not (black dots) (Fig. 5c). Based on the scatter plot, we divided genes into three groups and made a box plot to more clearly represent this (Fig. 5d and e). The y-axis of (Fig. 5e) represents the p-value for the comparison between gene expression levels at C14 and C14S1 in each group. The figures for H3K27me3 were moved to the supplemental file. We thank the reviewer for this suggestion because we think that the figures are now much easier to understand.

Comment 14

Figure 6 should be simplified

Suggestion for 6a: Venn diagram of me1-, me3, me2-marked loci and LDL3 (or only me1, me3 and LDL3 but need to justify this approach, currently not clear) eventually together with a, complementary PCA analysis with me1, me2, me3, LDL3 profiles

together with *ldl3* misregulated genes – this should show the components that better describe LDL3 targets

Suggestion for 6b: call clearly the plots “LDL3 targets associated with”: “H3K4me3 and H3K4me1”, “H3K4me1 only”, “H3K4me3 only”, “no H3K4 methylation” – but here again, I would suggest a scatter plot, or a different representation. Currently all the mean ratio of gene expression *ldl3*/WT are close to 1 thus does not highlight the misregulated genes.

If the authors want to keep box plots, they should consider to separate the LDL3 targets that are misregulated and analyse their preferential H3K4me1, me2 and me3 levels in WT and mutant.

Choose one clear message and place the rest in the sup data

Response

As suggested by the reviewer, we used a Venn diagram for Fig. 6a and scatter plots for Fig. 6b. We are grateful for these suggestions because the changes have helped us better convey a clear message. LDL3 preferentially bound to the genes with me1 and me3 modifications (Fig. 6a), and the time-lagged suppression of gene expression is observed in these genes (Fig. 6b and c).

Comment 15

L290-291 and later + figure 6: the choice to add a “+” or a “-“ to me1, me2, me3 to describe their absence/presence is not conventional and creates a complex setup for the text and the figure 6.

Response

This form of presentation was based on a study by Zhang *et al.*, (2009 Genome Biology) and other papers. We would like to keep this because it enables the sentences to be concise, and is simple to label the figure panels.

Comment 16

L271-280. The term bivalent modification is valid only for the case where H3K4me and H3K27me are found on the *same* nucleosome. This can be addressed only in sequential ChIP experiments, but not in the experimental set up done here

Response

We deleted “as a bivalent partner” from this sentence (line 298).

Comment 17

L289-L322 This paragraph is confusing. I suggest the authors to choose a different concept of presentation, perhaps starting with a PCA analysis (me1, me2, me3, LDL3 target, *ldl3* misregulated genes) to show which marks describe best the LDL3 targets, or if subgroups of genes are visible. The Venn diagram Figure 7a is good and comes

as a complement to the PCA.

Response

We introduced a Venn diagram as suggested, which we think improves the clarity of our conclusion (please also see the above response to comment 14).

Comment 18

L325-L335 I would drastically simplify the paragraph in 1-2 sentences (To identify LDL3 target genes involved in shoot regeneration, we selected 16 genes bound by LDL3, upregulated in wildtype upon shoot induction but downregulated in *ldl3* and hypermethylated on H3K4(me2) at the callus stage (Figure 7a).

Response

Because we need to explain why we focused on the very early stage of shoot induction (S1), unlike the previous studies, we think the first four sentences of this paragraph are necessary. Therefore, we keep these sentences as are in the previous version.

Comment 19

L345-354 belongs to discussion

Response

We moved this part to the Discussion (line 431–).

Comment 20

L375-376 “shows that H3K4me2 plays a role in transcriptional repression rather than activation” is a not well substantiated. The data do not show clearly a causality. Are high levels of H3K4me2 associated with low gene expression levels in wild-type? (the data focus on the fold changes in the mutant and wild type), do K4me2 levels increase on SIM as they do on CIM between C0 and C14 (Supp fig4) ? In fact The Supp fig5 show that high levels of K4me2 do not correlate with higher gene expression levels while high K4me3 levels do. But there is no indication that K4me2 represses gene expression. Instead, the fact that LDL3 target genes show lower expression in the mutant together with higher K4me2 suggests that the removal of this mark OR the stepwise methylation towards me3 state may be necessary to sustain high levels of transcription. How do the levels of K4me3 compare for these specific *ldl3* downregulated genes?

Response

K4me2 levels were unchanged following SIM treatment (Fig. 5a). K4me3 levels on LDL3-bound genes were not increased in the *ldl3* mutant compared with wt, while K4me2 levels were drastically increased (Supplemental Fig. 8a). Additionally, the alteration of K4me3 levels on LDL3-bound genes in *ldl3* did not correlate with changes in the expression of these genes in *ldl3* (Supplemental Fig. 8b). In contrast,

altered K4me2 levels in *ldl3* calli were negatively correlated with changes in gene expression in *ldl3* upon shoot induction (Supplemental Fig. 8b and Fig. 5). Thus, the stepwise methylation to me3 will not explain the phenomenon in this case.

Comment 21

L393-395 Please clarify the idea of stepwise histone methylation / DNA methylation-why DNA methylation?

Response

We originally included this because previous studies reported that DNA methylation shapes the histone modification landscape in other systems. However, because we did not describe these examples, we have deleted 'DNA methylation' from this sentence.

Comment 22

L405-406 "HAT might play roles", "it might be possible that".. it reads like overspeculation. Instead the authors could mention that in the light of this model in animal cells, future investigation should test possible interaction and co-occupation of HAT and LDL3 on target genes primed in callus and activated during shoot regeneration

Response

We revised the text according to the reviewer's suggestion (line 413-).

Comment 23

L409 "or more directly, another possible scenario" ◇ "a second, non exclusive scenario to be tested involves the recruitment of specific TF by LDL3 onto primed genes"

Response

We revised the text according to the reviewer's suggestion (line 415-).

Comment 24

L413-421: please reformulate for more concise and less speculative and/or describe a model to be tested ("however under this hypothesis [above], the delayed transcriptional activation remains to be explained. Whether phytohormone-responsive TF primed on target genes are involved remained to be determined")

Response

We revised the text according to the reviewer's suggestion (lines 418-423).

Comment 25

L425 why the answers to the above questions (which answers?) explain the absence of drastic phenotype in *ldl3* plants? What about functional redundancy, compensation by other factors regulating the homeostasy of H3K4me2? Alternatively, LDL3 expression just creates domains of competence which are dormant until the environment stimulates regeneration?

Response

We revised the text according to the reviewer's suggestion (line 442–). We also discussed other possibilities.

Reviewer #3

Comment 1

The authors claimed that LDL3 eliminates H3K4me2, however, the results from Suppl. Fig. 4A confirms that there is actually a very high increase in H3K4me1 and H3K4me3, which could be the responsible for such effect. By analyzing all the data presented (Suppl. Data 4B, Suppl. Data 5, Suppl. Data 6) all indicates that the conclusions that LDL3 works as a histone demethylation are not correct. How the authors can explain the abundant accumulation of H3K4me1 and me3 after callus induction in Suppl.

Response

The demethylase activity of LDL3, which mainly acts on H3K4me2, has been demonstrated in our new data (Fig. 4d and e).

We think that the high increase in H3K4me2 and me3 levels in wild-type explants during callus formation shown in Supplemental Fig. 5a (Supplemental Fig. 4a in the previous version) is not caused by LDL3 functions, but by other epigenetic regulators based on the following facts:

- H3K4me1 and me3 levels were rarely changed between wild-type and *ldl3* calli, but only H3K4me2 was increased in *ldl3* (Fig. 4a). The same tendency was observed even when focusing on LDL3-bound genes (new Supplemental Fig. 8a).
- Fig. 6a showed that LDL3 preferentially bound to genes with H3K4me1 and me3 modifications. Thus, as shown in Supplemental Fig. 5b, not only the levels of H3K4me2 but also those of H3K4me1 and H3K4me3 were detected as high on LDL3-bound genes in wild-type explants.

Thus, H3K4me2 appears to be increased (probably by some other molecule) but simultaneously eliminated by LDL3 during callus formation in wild-type explants.

Comment 2

Figure 4B. It is not clear to me why the authors only show three genes (Fig. 7E) to conclude their findings

Response

The three genes are shown as new regulators of *de novo* shoot regeneration. They respond to shoot induction at a very early stage of SIM incubation through H3K4me2 removal mediated by LDL3. In our assay, three out of 16 genes selected were involved in shoot regeneration, which we think strongly supports the proposed pathway (LDL3 – H3K4me2 removal in CIM – activation of the genes upon shoot induction – shoot regeneration).

Comment 3

Furthermore, It is not clear how the primed state is demonstrated in the results.

Response

Although we did not observe characteristic events for the primed state, such as de-condensation of the chromatin structure and time-lagged binding of RNA polymerase II to target genes, our data clearly demonstrated a correlation of H3K4me2 levels with gene expression states after shoot induction, but not with those in the same stage (Fig. 5 and Supplemental Fig. 8). The data also demonstrated that the effects are caused through LDL3-mediated H3K4me2 removal on target genes (Figs. 4, 5, Supplemental Figs. 5 and 8). These results indicated that H3K4me2 states are regulated by LDL3 in calli primed for gene expression states, which are activated by shoot induction.

Comment 4

Page 5 lines 82-84: The authors stated that "in many plant species and tissue types *de novo* organogenesis is only achieved from pre-induced callus, and not directly from native tissue" However, the cited reference does not state that information. There is plenty of published information about the *Novo* callus formation (Book Plant Cell Culture Protocols, Springer 2018 <https://link.springer.com/book/10.1007/978-1-4939-8594-4>). Also, the authors need to consider the habituation effect in some cultures in order to make a better discussion.

Response

We cited more suitable literature in line 85.

We think that the acquisition of regenerative competency by callus cells and the habituation effect on them are different phenomena.

The long incubation of explants on CIM does not always provide regenerative competency to the cells, although a certain period is necessary as described in the text.

The callus cells usually lose their regenerative competency by overly long CIM incubation. In contrast, the habituation effect is often caused by continuous culture, and its effect is heritable. Although we cannot exclude the possibility that both processes share common molecular mechanisms, it is difficult to link them at this point. Therefore, we did not discuss habituation effects in this manuscript.

Comment 5

Page 6 line 103: The authors stated that "Few reports have described the epigenetic basis of the acquisition of shoot regenerative.." however, CIM and SIM studies have been investigated from an epigenetic point of view. (Stroud H, Ding B, Simon SA, Feng S, Bellizzi M, Pellegrini M, Wang G-L, Meyers BC & Jacobsen SE (2013) Plants regenerated from tissue culture contain stable epigenome changes in rice. eLife 2 doi:10.7554/eLife.00354. Vining K, Pomraning K, Wilhelm L, Ma C, Pellegrini M, Di Y, Mockler T, Freitag M & Strauss S (2013) Methylome reorganization during in vitro dedifferentiation and regeneration of *Populus trichocarpa*. BMC Plant Biology 13(1):92)

Response

Although genome-wide epigenetic modifications have been described in the context of *de novo* organogenesis (CIM and SIM studies) as stated by the reviewer, few studies identified a specific epigenetic modification that directly regulates the acquisition of shoot regenerative competency. The papers suggested by the reviewer reported whole genome DNA methylation profiling at several time points of the regeneration process. Although these are very informative, they did not demonstrate whether particular DNA methylation on certain regions of the genome is important for the acquisition of regenerative competency. Moreover, because we are specifically referring to the molecular mechanisms underlying the acquisition of competency, we did not refer to these papers here.

Comment 6

Figure 8 seems quite deterministic because it shows as a unique player for the acquisition of shoot regenerative competency to LDL3 and the effect of demethylation in H3K4me2. However, as I said, with the presented results, it could be due to the hypermethylation in H3K4me1 and me3. I think that more experiments need to be done in order to confirm that 1) the acquisition of shoot competency is due to H3K4me2 and not by H3K4me1 or me3; 2) that LDL3 is indeed a demethylating gene and not a hypermethylating player; 3) that there is a priming mechanism behind the process.

Response

1) This is related to comment 1 (please see above).

Compared with the wild-type callus, only H3K4me2 among the three H3K4

methyations showed up-regulation at LDL3-bound genes in the *ldl3* callus (Supplemental Fig. 8a).

We also examined the relationship between hyper-H3K4 methylations caused by *ldl3* in calli, and the altered gene expression of mutant explants before and after shoot induction at LDL3-bound genes (Supplemental Fig. 8b). The results indicate that hyper-H3K4me2, but neither hyper-H3K4me1 nor -H3K4me3, caused by *ldl3* in the callus affects later gene activation upon shoot induction, but not during callus formation. Moreover, LDL3 preferentially demethylates H3K4me2 (Fig. 4).

Taken together, we conclude that the loss of shoot regenerative competency in *ldl3* is caused by H3K4me2, but not H3K4me1 or H3K4me3.

- 2) The demethylase activity of LDL3, which acts on H3K4me2 and H3K4me3, has been demonstrated in our new data (Fig. 4d and e).
- 3) This is related to comment 13 (please see above).

Our data demonstrated a correlation of H3K4me2 levels with gene expression states after shoot induction, but not with those in the same stage (Fig. 5 and Supplemental Fig. 8). They also demonstrated that the effects are caused through LDL3-mediated H3K4me2 removal on target genes (Figs. 4, 5, Supplemental Figs. 5 and 8).

These results indicated the presence of gene priming mechanisms that are regulated by LDL3-mediated H3K4me2 removal. Future work should determine how the reduced H3K4me2 states prime the gene expression states.

REVIEWERS' COMMENTS:

Reviewer #1 (Remarks to the Author):

The authors have addressed most of my concerns. I have no further suggestions.

Reviewer #2 (Remarks to the Author):

I acknowledged that the authors have extensively revised the manuscript, profoundly improving text and figure presentation. The additional experiment also suggested by Reviewer 1 and provided in the revised version is a convincing dataset strengthening the conclusions of the work.

Reviewer #3 (Remarks to the Author):

I have read the answers to my questions and the changes made in the new version of the manuscript. I have not further comments to make and the manuscript is much better now.

Point-by-point letter to reviewers' comments and editorial requests

Reviewer #1 (Remarks to the Author): The authors have addressed most of my concerns. I have no further suggestions. Response: Thank you for approving our revised manuscript.

Reviewer #2 (Remarks to the Author): I acknowledged that the authors have extensively revised the manuscript, profoundly improving text and figure presentation. The additional experiment also suggested by Reviewer 1 and provided in the revised version is a convincing dataset strengthening the conclusions of the work.

Response: Thank you for approving our revised manuscript.

Reviewer #3 (Remarks to the Author): I have read the answers to my questions and the changes made in the new version of the manuscript. I have not further comments to make and the manuscript is much better now.

Response: Thank you for approving our revised manuscript. We carefully revised our manuscript according to your